# Molecular Understanding and Practical In Silico Catalyst Design in Computational Organocatalysis and Phase Transfer Catalysis—Challenges and Opportunities

**DOI:** 10.3390/molecules28041715

**Published:** 2023-02-10

**Authors:** Choon Wee Kee

**Affiliations:** Institute of Sustainability for Chemicals, Energy and Environment (ISCE2), Agency for Science, Technology and Research (A*STAR), 1 Pesek Road, Jurong Island, Singapore 627833, Republic of Singapore; kee_choon_wee@isce2.a-star.edu.sg

**Keywords:** organocatalysis, phase transfer catalysis, DFT, machine learning, organic reactions, solvation, potential energy, computational chemistry, free energy, kinetics

## Abstract

Through the lens of organocatalysis and phase transfer catalysis, we will examine the key components to calculate or predict catalysis-performance metrics, such as turnover frequency and measurement of stereoselectivity, via computational chemistry. The state-of-the-art tools available to calculate potential energy and, consequently, free energy, together with their caveats, will be discussed via examples from the literature. Through various examples from organocatalysis and phase transfer catalysis, we will highlight the challenges related to the mechanism, transition state theory, and solvation involved in translating calculated barriers to the turnover frequency or a metric of stereoselectivity. Examples in the literature that validated their theoretical models will be showcased. Lastly, the relevance and opportunity afforded by machine learning will be discussed.

## 1. Introduction

Organocatalysis’s contribution to modern society has been recognized by one of the pinnacle scientific awards—the Nobel Prize. Tremendous advancement in catalyst design and activation mode has been achieved since its inception [1,2]. Phase transfer catalysis (PTC) usually involves using a lipophilic organocatalyst to facilitate the movement of reactive ions into an organic phase where the reactants are in high concentration [3]. Its merits include the potential to use extremely low catalyst loading and to provide access to the reactivity of otherwise insoluble ions. The centerpiece of asymmetric PTC is tight ion-pairing. Cationic chiral catalysts such as quaternary ammonium-based [4,5], guanidinium-related [6], and phosphonium-related [7] salts have been extensively reported. Gouverneur and co-workers reported the use of neutral urea or thiourea as PTC in the activation of inorganic fluorides [8]. Phosphate anions enable reversed-polarity phase transfer catalysis where the chirality originates from the anionic component of the catalyst [9]. PTC adds another dimension of complexity to understanding the reaction mechanism and in silico prediction, as the reactions could occur at the interface (liquid/liquid or solid/liquid) [10].

Various reviews on or with emphasis on computational organocatalysis include the works from the group of Duarte [11], Krenske [12], and Trujillo [13,14]. Modeling with computational chemistry provides microscopic insight at a level that cannot be otherwise obtained experimentally. With continued advancement in hardware and computational chemistry software, increasingly more realistic and accurate simulations can be performed. For instance, the time taken to compute the energy of a molecule with a given size in the 2008 commentary of Houk and Cheong has substantially reduced due to these advances [15]. From an organocatalysis or PTC developmental point of view, a balance between accuracy and time required is of paramount importance. Most modern organocatalytic and phase transfer (PT) catalytic systems have sufficient complexity (in terms of mechanism and conformational degree of freedoms) that often render in silico prediction of catalytic performance (e.g., turnover frequency: TOF and level of stereoselectivity [16,17]) impractical relative to empirical or domain expertise-driven experimentation. Practical is a relative term that depends on the computational resources available to the research group. However, it is reasonable to assume that most research groups would not have exclusive access to computing resources equivalent to the one used to generate the open-catalyst project data [18,19]. As such, continued synergetic development in hardware and software is still required to reach a practical in silico catalyst design accessible to the average research group. Pertinent to this point, readers are encouraged to refer to the excellent outlooks on the future of computational chemistry in the perspectives of Grimme and Schreiner [20], Houk and Liu [21], and Sunoj [22].

In this review, we will elaborate on the main points in the succinct overview by Harvery et al. [23], and Funes-Ardoiz and Schoenebeck [24], but in the context of organocatalysis and PTC. We will highlight the challenges and opportunities in computational organocatalysis and PTC from both the reactivity and the selectivity perspective. While we will prioritize more recent work (2018 and beyond), classic examples that are representative of a particular key aspect of computational organocatalysis or PTC will be discussed even if it predates our arbitrary definition of recent. Our approach will be balanced—we will provide examples demonstrating notable advances and caveats based on the literature. Proof-of-concept calculations will be included in this review if they reinforce essential concepts.

### 1.1. Overview

The flow of this review will follow a typical workflow to study a reaction via the location of key stationary points on the potential energy surface or PES (Figure 1); this is currently the dominant workflow for computational organocatalysis or PTC, as the alternative—to study each elementary step via molecular dynamics (coupled with enhanced sampling) is currently not practical or even intractable in a realistic model. The numbering in Figure 1 corresponds to the sequence in which the mentioned topic will appear in this review. The readers can refer to the next Section 1.2 Content for the general structure of this review.

### 1.2. Content

Section 2.1.1. Geometry Optimization: The location of key stationary points requires at least the capability to calculate the energy and gradient (first derivative of energy with respect to coordinates) of a given structure; therefore, we will first discuss the choice of method to obtain these quantities, and its impact on quality based on theoretical benchmarks (1 in Figure 1). Density functional theory (DFT) has emerged as the method of choice to obtain these key stationary points (equilibrium geometries and transition state structures) on the PES.

Section 2.1.2. Compound Method or “Double-Barreling”: Upon obtaining these key stationary points, it is a common practice to calculate the electronic energy at a different level of theory (2 in Figure 1). Extensive theoretical benchmarks of density functional approximation (DFA) are based on this concept. We will provide the recommended DFA obtained from a selection of gas-phase benchmark studies. Works that explore the limit of current theoretical methods will be highlighted as they are important for theoretical development. In addition, they caution the applied computational chemists to potential pitfalls. The compound method provides the opportunity to include the influence of solvent via an implicit solvation model. However, due to the challenges associated with modeling the solvent, we will defer a more detailed discussion to a later section.

Section 2.2. Thermochemical Corrections: Standard statistical thermodynamics equations are usually employed to obtain key experimental observables such as Gibbs free energy (3 in Figure 1). Theoretically, the PES surface is not an experimental observable; a physically accurate comparison requires calculating the free energy. 

Section 2.3. Conformational Sampling: With the capability to calculate the free energy of a given molecular structure, conformation sampling is discussed with emphasis on its importance and the software available to achieve automated sampling (4 in Figure 1). 

Section 2.4. Translating from Calculated Barrier to Rate Constant and Section 2.5. Extending to Catalysis—Turnover Frequency: Once all the relevant stationary points and their respectively thermally accessible conformers are obtained, translation from the energy representation to the rate representation will be required (5 in Figure 1). Transition state theory (TST) bridges the theoretically derived barrier and experimental rate and, consequently, reaction performance metrics such as selectivity and turnover frequency (TOF). 

Section 2.6. The Computational Challenge—Mechanism: We will highlight some challenges associated with investigating key stationary points on the PES. For instance, the role of essential additives or reagents may not be obvious, and the potential of aggregation of catalysts (Section 2.6.1 Non-Linear Effect in Asymmetric Catalysis) further complicates the matter. Theoretical considerations will be examined here, such as the assumption of one TS structure leading to multiple products due to dynamic effect (Section 2.6.2 Bifurcating potential energy surface).

Section 2.7. Modeling Solvation: Solvents are central to organocatalysts and PTC. Their influence on the reaction outcome could be potentially game-changing (6 in Figure 1). This section considers the various theoretical approaches to model the solvent’s influence. Examples from the literature with emphasis on solvation will be discussed in Section 2.7.1 Examples Related to Solvation.

Section 2.8. Validation of Calculations: With the challenges and background discussion set, the audience would have realized the numerous assumptions in obtaining key stationary points of the catalytic reaction. We will showcase some examples where the models (combination of stationary points and level of theory) are validated by their capability to predict several experimental data points (e.g., kinetic isotope effect and enantiomeric excess).

Section 2.9. Understanding/Interpretation: An analysis to propose the origin of selectivity or reactivity is appealing to experimental chemists, as it could help augment their domain expertise and inspire the development of new catalysts based on an updated chemist’s perspective toward the reaction in question.

Section 2.10. Machine Learning and its Relevance to the Field: With the advent of artificial intelligence and machine learning, many opportunities have arisen to tackle the challenge of in silico catalyst design differently. We will highlight examples that map descriptors derived from molecular structures to various performance metrics (Section 2.10.1. Supervised Learning) and 3D structures to potential energy and forces (Section 2.10.2. Machine Learning Potential).

## 2. Discussion

### 2.1. Choice of Methods

The method of choice for most computational studies that involve organocatalysis or PTC is density functional theory (DFT). The researcher is immediately faced with choosing which density functional approximation (DFA) to use. Benchmark studies in the literature offer valuable guidance on the performance of various DFA on geometry optimization, reaction barriers, and non-covalent interactions. It should be noted that these benchmarks are usually for calculations in the gas phase. Extrapolation and application to the condensed phase, where organocatalysis and PTC generally occur, requires careful validation by the computational chemist.

#### 2.1.1. Geometry Optimization

Obtaining a reasonable geometry is the prerequisite for calculating many properties of interest to a chemist. A geometry optimization involves searching the potential energy surface (PES) to find a stationary point. Analysis of the Hessian matrix reveals whether this stationary point corresponds to an equilibrium geometry (no imaginary frequency, reactant molecules, stable intermediate, or complexes in a reaction) or a transition state structure (one imaginary frequency). It forms the basis for studying a reaction mechanism via the location of stationary points on the PES.

Within the single reference wavefunction limit, coupled-cluster singles, doubles, and perturbative triples or CCSD(T) [25] at complete basis set (CBS) is generally accepted as the “gold standard” which can achieve an error of less than 1 kcal/mol (for chemical reaction calculation). However, geometry optimized at this level of theory is rare. Vermeeren et al. optimized key stationary points of five pericyclic reactions at CCSD(T)/cc-pVTZ [26]. They benchmarked the BP86 gradient corrected functional with basis sets of various sizes and the use of empirical dispersion correction. They found that BP86 with a double zeta basis set gave the smallest transitional and rotational projected Root-Mean-Square Deviation (RMSD). A larger basis set worsens the RMSD relative to the CCSD(T) reference, and dispersion correction improvement is inconsistent amongst all the stationary points tested. In the benchmark study of various ionic liquid clusters by Seeger and Izgorodina, ωB97X-D/aVDZ and M06-2X were recommended by the authors based on FMO3-SRS-MP2/cc-pVTZ geometries [27]. However, they emphasized that empirical dispersion correction is crucial.

The location of TS structures is usually the costliest part of a computational mechanistic investigation via the location of stationary points. Multiple exact Hessian calculations are typically needed for the optimization to be successful. DFT is usually the method of choice due to its computational cost and the availability of analytical Hessian, which may be essential in the successful location of TS structures. For relatively large systems such as modern organocatalysts and PTC, the choice of DFA and basis set available is limited. The use of a relatively less expensive generalized gradient approximation (GGA) functional such as PBE, BP86, and B97-D is common. Theoretical benchmark against structure from a higher level of theory is generally infeasible. Instead, the quality is usually assessed by whether experimental results can be reproduced. A non-exhaustive list of DFA/basis set combinations reported in the literature to optimize TS structures for various classes of asymmetric organocatalysts (we included one example of cooperative catalysis by Sunoj and co-workers [28]) is given in Figure 2. The Minnesota functionals and B3LYP appear to be the DFA of choice for different groups. For a very fast elementary step (barriers of a few kcal/mol), the choice of DFT and basis set can profoundly impact whether the TS could be located. For instance, Paton, Gouverneur, and co-workers found that the fluoride ion must be described with a sufficiently large and diffuse basis set, or a TS structure could not be located [29].

Bistoni and co-workers evaluated the impact of changing the DFA, basis set, and the inclusion of solvent on the RMSD of one of their TS structures (calculated at PBE-D3/def2-SVP) [40]. They found that with the def2-SVP basis set, the TPSSh-D3 and PBE-D3 gave an RMSD of 0.071 (hybrid vs. GGA DFA). The impact of increasing the basis set size—from def2-SVP to def2-TZVP(-f)—is more significant, giving an RMSD of 0.439. At the same time, the inclusion of solvent via the CPCM model only provides an RMSD of 0.043. The impact of these deviations in the predictive capability remained to be determined.

Geometric counterpoise correction by Grimme and co-workers includes an empirical correction to account for the effect of a finite basis set on geometry and vibrational frequencies. For DFT, PBEh-3c [41], R^2^SCAN-3c [42], and B97-3c [43] are available for different needs. Head-Cordon and co-workers proposed the DFT-C scheme to correct the basis set superposition error for the def2-SVPD basis set [44]. These corrections can be applied to a large system where geometry optimization and Hessian calculations with a larger basis set would be computationally too demanding.

The hybrid QMQM or the ONIOM [45] method can substantially speed up the location of key stationary points. However, their performance has not been extensively benchmarked or validated and may likely be system-dependent, although they could serve as starting points for further filtering or refinement of a large ensemble of conformers.

#### 2.1.2. Compound Method or “Double-Barreling”

It is a common practice to perform an additional single-point calculation on the optimized structure with a larger basis set or with an altogether different method, also known as “double-barrelling” or a compound method. The motivation can be to use a larger basis set with the hope of obtaining more accurate electronic energy, to include the solvent effect via an implicit solvation model, or to use a linear-scaling CCSD(T) in the hope of approaching chemical accuracy (<1 kcal/mol).

Various comprehensive benchmark studies based on this concept have provided valuable information on popular density functional approximation (DFA) performance in predicting different properties. Goerigk and co-workers performed extensive DFA benchmarks on the GMTKN55 dataset [46] (a collection of high-quality references for main-group thermochemistry, kinetics, and noncovalent interactions) [47,48,49]. The comprehensive benchmark by Mardirossian and Head-Gordon is an authoritative reference for the performances of over 200 DFA over a wide range of scenarios (noncovalent interaction, thermochemical data, barrier heights) [50].

Jacob’s ladder analogy [51] is used to classify DFA according to improvement from the lowest ladder rung (LDA: local density approximation). In LDA, the electron density is assumed to be uniform throughout space. Its uses in organocatalysis or PTC are expected to be limited. The addition of information on the electron density gradient (gradient Laplacian) brings us to the second rung—generalized gradient approximation (GGA). The non-separable gradient approximation (NGA), in which the exact and correlation are combined, proposed by Peverati and Truhlar, is classified under the second rung [52]. Meta-GGA/NGA at the third rung includes the kinetic energy density on top of the electron density and the density gradient. It represents the highest rung for semi-local DFA and provides good accuracy for equilibrium geometries [53]. The fourth rung is where computational organocatalysis work is generally utilized (see Figure 2). At the fourth rung, Hybrid GGA/ meta-GGA DFA replaces a specific degree of exchange in semi-local DFT with Fock exchange (occupied orbitals information). Finally, virtual orbital information is included at the fifth rung to form the double-hybrid DFA. The high cost of double-hybrid DFA restricts its use in organocatalysis or PTC to mainly single-point calculations.

We compiled a recommendation in Table 1 based on the cited references. The basis set is not listed, but they are generally at least triple-zeta quality. The reader should refer to the original literature for detailed technical information. The emphasis will be on DFAs and their performance in calculating barrier heights. However, the reader should note that accurate reaction thermodynamics is essential to predict metrics such as TOF. A lower value in Table 1 indicates better performance (within the benchmark dataset used).

The more computationally economical GGA generally produces a considerable error in calculating gas-phase barrier height (BH). In general, hybrid DFA has been consistently found to be the best performer in BH, except for B3LYP, which is a piece of common knowledge in the literature. The range-separated hybrid meta-GGA ω97M-V is generally recommended by various studies, as indicated in Table 1. Ho and co-workers’ benchmark on ionic S_N_2 reaction against canonical CCSD(T)/CBS demonstrates that its performance is impressive even with a small ma-def2-SVP basis set [54]. For the set of three reactions, they found that DSD-PBEP86-D3(BJ) tends to overestimate the barriers while ω97X-V tends to underestimate them. Their results indicate that the conclusion is reasonably independent of basis set sizes.

**Table 1 molecules-28-01715-t001:** Selected compilation of recommended DFAs at various rungs of the DFA Jacob’s ladder and their associated deviations in gas phase benchmark for barrier heights and noncovalent interactions.

Rung on the Jacob Ladder	DFA	Correction (Recommended by the Ref) ^[a]^	Barrier Heights			GMTKN55
			RMSD ^[b]^	WTMAD-2 ^[c]^	BH9 RMSD ^[d]^ (MAE ^[e]^)	WTMAD-2
**2: GGA/** **NGA**	B97	D3(BJ) [46]	8.32	13.15	11.27	8.55 [46]
		D3(0) [50]	8.65	NA	NA	NA
	revPBE	D3(BJ) [46]	8.30	15.79	11.24	8.27 [46]
		D3(0) [50]	8.26	NA	NA	NA
	BLYP	D3(0) [27,50]	10.13	NA	NA	NA
		D3(BJ) [46]	9.91	NA	12.10	9.51
**3: Meta-GGA/** **NGA**	SCAN	D3(BJ) [46]	7.83	14.94	NA	7.86 [46]
	r^2^SCAN	None	NA	NA	7.90	NA
		D3BJ	NA	NA	9.25	NA
		3c [42]	NA	NA	NA	7.5 [42]
	revTPSS	D3(BJ) [46]	NA	15.78	NA	8.50 [46]
	M06L	D3(0) [46,50]	6.84	7.56	NA	8.61 [46]
	B97M	V [48,50]	4.35	7.53	6.70 (4.14)	5.46 [48]
**4: Hybrid GGA/Meta-GGA**	ωB97M	V [48,50,54,55,56]	1.68	3.40 ^[f]^	(2.08)	3.53 [48]
	ωB97X	V [46,50,56]	2.44	4.21 ^[f]^	(3.20)	3.98 [46]
		D3(0) [46,50]	2.28	4.67	NA	4.61 [46]
	M06-2X	None [56]	NA	NA	(2.27)	NA
		D3(0) [26,50]	2.60	5.60	NA	4.94 [46]
	B3LYP-D3	D3(0)	5.92	NA	NA	NA
		D3BJ	NA	NA	6.77	6.42 [46]
**5: Double-Hybrid**	DSD-PBEP86	D3(BJ) [46]	NA	3.52	4.04	3.14 [46]
		NL [48]	NA	3.25	NA	2.84 [48]
	revDSD-PBEP86	D3(BJ) [26,55]	NA	NA	2.96	2.42 [55]
	ωB97X-2	D3(BJ) [48]	NA	3.25	NA	2.97 [48]
	ωB97M(2)	None [55]	NA	NA	NA	2.19 [55]
	DSD-BLYP	D3(BJ) [46]	NA	3.04	NA	3.08 [46]
		NL [48]	NA	2.86	NA	3.05 [48]
	B2GPPLYP	D3(BJ) [46]	NA	3.24	NA	3.26 [46]

NA: Not Available. ^[a]^ Empirical correction (mostly dispersion). References that recommend this DFA + correction combination are indicated by the citation(s). ^[b]^ RMSD: Root Mean Squared Deviation in kcal/mol from ref [50]. ^[c]^ WTMAD-2: Weighted Total Mean Absolute Deviation 2 in kcal/mol. See ref [46] for the definition. ^[d]^ RMSD based on the entire BH9 dataset [56]. Values in kcal/mol from ref [57]. ^[e]^ Mean Absolute Error of the entire BH9 dataset. Values in kcal/mol from ref [56]. ^[f]^ post-SCF version of non-local correction.

Double hybrid functional, which currently represents the highest rung of the DFT Jacob’s ladder, is generally the most reliable from gas phase benchmark results. Goergik and coworkers reported a double hybrid benchmark of the GMTKN55 dataset [58]. The group of Martin has performed extensive work on double hybrid DFA [55,59,60]. They have reported the impact of basis set size for double-hybrid DFA and the use of F12 explicit correlation to accelerate basis set convergence [61]. For double-hybrid DFA, a non-local post-SCF correction was beneficial to both barrier heights and the overall noncovalent interaction dataset in the GMTKN55 benchmark [48].

The domain-based local pair natural orbital (DLPNO [62,63]) and the localized molecular orbital (LMO [64,65]) approximations, implemented in ORCA [66] and MRCC [67], respectively, have enabled single-point calculation to be performed on a molecule with around up to a few thousand atoms (or tens of thousands of basis functions) [65,68]. These generally approximate the canonical CCSD(T) result for large molecules in the benchmark dataset described above. Bistoni and co-workers have reported the DLPNO-CCSD(T)/def2-TZVP single-point calculations for two computational studies on organocatalysis [37,40].

The accuracy of DLPNO and LNO approximations are benchmarked against a small system where the corresponding canonical CCSD(T) calculation is tractable. For DLPNO-CCSD(T), its accuracy is dictated by three main cutoffs to determine the molecular orbital space that will be included. For the recommendation of cutoff choice and its impact, the reader is referred to the work of Neese and co-workers [69] and Seeger and Izgorodina (on ionic liquids) [70]. In their ionic S_N_2 computational study, Ho and co-workers presented a comparison of canonical and DLPNO CCSD(T) with various basis set sizes [54]. For related considerations, and different schemes that utilized DLPNO to approximate the canonical result, the readers are encouraged to refer to the references cited in this paragraph and the references cited within them.

Brandenburg, Tkatchenko, and co-workers reported that for large molecules (up to 132 atoms), the disagreement between LNO-CCSD(T)/CBS and the quantum diffusion Monte Carlo (DMC)-calculated interaction energy can be of the order of 7.6 kcal/mol (Figure 3) [71]. 

Based on the structures provided by Brandenburg, Tkatchenko, and co-workers, we evaluated the interaction energies from a selection of recommended DFA, as given in Table 1. It is reasonable to assume that dispersion interaction will dominate the interaction between C60 and 6-CPPA. The significance of dispersion is especially evident in the B3LYP/def2-TZVPP calculations in Figure 3. Without any empirical dispersion correction, the interaction between C_60_ and 6-CPPA is repulsive. The choice of dispersion correction will significantly determine if the result is closer to the LNO-CCSD(T) or the FN-DMC values. M06-2X, ωB97X-V, and DF-RPA are the closest to the LNO-CCSD(T) values. In the following paragraph, we emphasize that it is not an exhaustive study but a quick demonstration that may have significance in the double-barreling discussion. The problem of approximate solution to CCSD(T) of a large system where dispersion dominated the system (which is very well described) has been pointed out by the studies of Herbert and co-workers [72] and Patel and Wilson [73]. Lastly, Hobza and co-workers pointed out that CCSD[T] is superior to CCSD(T) in describing noncovalent interaction [74].

The caveat on double-barrelling has been discussed by Kevin, where he analyzed the error introduced in double-barrelling by varying the bond lengths of small molecules about their equilibrium positions [75]. The C_60_ and 6-CPPA host–guest complex will be used to illustrate this point further. With B3LYP/6-31G(d) optimized geometry, the complex is a transition state (TS) structure which corresponds to the expulsion of C_60_ from the complex (ref to the blue arrows in Figure 3). Single-point calculations with empirical dispersion correction indicate that their interaction is attractive, which contradicts it being a TS that expels C_60_ from 6-CPPA.

Pertinent to organocatalysis, Chin and Krenske highlighted a similar problem of double-barrelling in their highly detailed computational study of chiral phosphoric acid (CPA)-catalyzed asymmetric Nazarov reaction. The calculated % *ee* varies from −24% to +94% with various modern DFA, basis set, and implicit solvation models [76]. Despite this and the challenges highlighted before, we wish to emphasize that a successful prediction of experimental enantioselectivity is unlikely not to have an element of error cancellation to various and often unknown extents. This brings us to testing the computational methodology against diverse experiment data points to increase confidence in the calculated results (see Section 2.8 Validation of Calculations).

### 2.2. Thermochemical Corrections

Under most organocatalytic reaction conditions, the most relevant quantity to predict reaction rates (consequently yield) and selectivity (enantio- or diastereo-) is Gibbs free energy, as most reactions are performed under constant pressure (usually atmospheric). 

The statistical thermodynamics approach is then used to calculate the correction for Gibbs free energy, which is then added to the electronic energy to obtain the Gibbs free energy of a molecule. The assumptions in deriving the equations apply—ideal gas, rigid rotor, and standard pressure. The impact of approximating low vibrational frequencies as a rigid rotor is well documented. Various corrections are available to handle the correction of these low vibrational frequencies [77,78,79]. The open-source PYTHON code GoodVibes from Paton’s group allows different schemes of correction to the rigid rotor to be applied as a post-processing step [80,81].

Plata and Singleton highlighted the importance of various important but widely ignored corrections to the entropy calculated with standard statistical thermodynamic equations [82]. These include residual entropy due to symmetry and entropy of mixing.

Thermochemical correction when solvation is involved is an area of extensive debate. The controversy lies in how to handle translational and rotational entropy. The approximation implemented in most electronic structure packages is the ideal gas/rigid rotor/harmonic oscillator (RRHO) approach. Some authors have shown that the RRHO can be used without correction in selected cases [83,84,85,86]. In contrast, others recommend various corrections (the reader is referred to the work of Besora et al. [87] for details of all these corrections). These corrections are evaluated by Besora et al. by comparing values derived from umbrella sampling with explicit solvation (molecules of solvation are fully represented at the atomic level) [87]. Jacobsen and co-workers tested RRHO and two quasi-RRHO schemes based on calculated levels of enantioselectivity vs. experimental ones (for more detail, please refer to Section 2.8 Validation of Calculations) [88].

### 2.3. Conformational Sampling

Except in the extreme cases where the molecules or catalysts have minimal flexibility (or rotatable bonds), a low-quality ensemble of conformers can result in biased results. In calculating the contribution of competing pathways (Blue vs. Red in Figure 4), the Curtin–Hammett principle is usually assumed to be valid—the relative free energies of the appropriate ensembles of TS structures can be used for prediction contribution from each competing pathway. With this assumption fulfilled, one can utilize the Boltzmann distribution to calculate a selectivity metric. Selectivity, in this case, can be relative product distribution, diastereoselectivity, or enantioselectivity. For enantioselectivity, the tutorial review by Paton and co-workers provides the relevant details and equations to perform this calculation [16]; the same concept can be used for other forms of selectivity listed.

Figure 4A shows a hypothetical situation where only one TS structure for each pathway was located. This is an extreme case, in which for each pathway, it is sufficient to consider only one TS as the other conformations’ contribution is negligible due to their high energies. However, in this scenario, the lowest energy TS_blue_ is not located. Therefore, P_red_ is predicted to be the primary product. There is a possibility that a lack of conformation sampling can lead to a fortuitous agreement with the experimental result. However, extensive validation with multiple experimental outcomes is expected to allude to this deficiency. In Figure 4B, an ensemble of TS structures must be considered for each pathway, as many conformations contribute significantly to product formation. The standard procedure is to calculate the product selectivity based on a Boltzmann population derived from the relative energies of the two combined ensembles.

As discussed above, critical conformations (those with significant Boltzmann populations at the reaction temperature and pressure) must be located. This step is usually done automatically by using conformational sampling software. The conformational space search algorithm and the accuracy of the Hamiltonian employed to evaluate the energies during the search are two critical aspects for a successful conformation sampling. However, many structures are usually generated during the search, and each must be optimized. Therefore, DFT-based methods are generally infeasible. The use of classical force field is common but risky, as they are usually not trained on the system of interest; it risks missing key conformations if an energy-based cut-off is employed. As a compromise, semi-empirical methods or potential Machine Learning Potential (vide infra) can be used. The open-source CREST, coupled with the GFN2-XTB tight-binding Hamiltonian, is a readily available and powerful tool for this purpose [89]. Its non-covalent interaction sampling mode is particularly suited for sampling TS structures, as key internal coordinates of the TS can be restrained during the sampling.

For an evaluation of various conformational sampling methods in quaternary ammonium-based PTC and the application of unsupervised machine learning techniques for analysis of the results, the reader is referred to the work of Iribarren and Trujillo [90].

### 2.4. Translating from Calculated Barrier to Rate Constant

In a chemical reaction with one elementary step, such as cycloaddition, and nucleophilic substitution, the rate constant and the concentration(s) of the reactant, at any point in time, determines the rate of product formation. If a pre-equilibrium kinetic model is assumed—an equilibrium exists between the reactive conformation (or RC: reactant complex) and the reactant(s) in their most stable state—then the effective rate constant (*k*K_pre-equi_) can be obtained from the free energies of the TS and reactants. Our work in the in silico design of halogen bond-based catalysts uses this model to compare with the predicted TOF of our proposed catalysts [91].

The conventional transition state theory (TST) links theoretical activation energy (ΔG^‡^ in Figure 5) and the experimental rate constant. A more sophisticated form of the TST, such as the variational TST, is expected to improve the accuracy of the rate constants [92]. However, their required computational costs are generally intractable for system sizes found in most organocatalysis.

Tunneling, a quantum mechanical phenomenon, can result in a larger rate constant than predicted from the barrier height. The semiclassical Wigner correction provides a simple means to determine the importance of tunneling (Equation (1)) [93]. However, Truhlar and co-workers recommended using this correction if κ(T) does not exceed 1.2 due to the various approximations involved [94]. The open-source pyQuiver can calculate tunneling correction via the Wigner and Bell inverted parabola from a cartesian Hessian matrix [95].
(1)$$\kappa \left(T\right)=1+\frac{1}{24}{\left|\frac{h\omega^{‡}}{2\pi k_{B}T}\right|}^{2}$$

Experimentally, the involvement of quantum tunneling effect has been reported in the seminal work of Schreiner and co-workers where they introduced the concept of a tunneling-controlled reaction [96], in addition to the commonly used kinetic- and thermodynamic-controlled reaction [97]. More sophisticated tunneling correction exists, but sophistication generally comes with an increasing computational cost [98].

Diffusion-controlled rate constant approximation, such as the equilibrium between reactant(s) and RC, can be calculated with Equation (2). r_A_ and r_B_ are radii of the reactant A and B, respectively, which can be estimated from an intrinsic solvation model calculation. The procedure to calculate W, D_A_, and D_B_ is detailed in the work of Pliego and co-workers [99].
(2)$$k_{diffusion}=\frac{4\pi \left(r_{A}+r_{B}\right)\left(D_{A}+D_{B}\right)W}{e^{W}-1}$$

### 2.5. Extending to Catalysis—Turnover Frequency

In catalysis with a multi-elementary steps cycle with potential non-productive branches, turnover frequency (TOF) is frequently employed to measure the efficiency and stability of a catalyst. A decreasing TOF over time indicates catalyst deactivation as the reaction progresses. The prediction of TOF will be more challenging as the structure difference between the transition state and reactant(s) would translate to vastly different electronic structures and correspondingly less favorable error cancellation. Bearing in mind the exponential relationship between rate constants and barrier height, a 1 kcal/mol difference (upper bound of chemical accuracy) may translate to up to an order of magnitude difference in TOF (Table 2). For catalyst design, however, one might be more concerned with relative TOF between different catalysts than the absolute TOF of a single catalyst.

A catalytic cycle is a sequence of interconnecting elemental steps described by a set of coupled ordinary differential equations (ODE). Obtaining an analytical expression is usually not possible for complicated catalytic cycles. These ODE can be solved numerically to high precision with various ODE solvers (Figure 6). However, when the rate constants are vastly different in magnitudes, the time step for the numerical integration must be sufficiently small to ensure the stability of the simulation. As a result, the time scale that can be transverse could be in the picosecond region or less. This is highlighted and tackled in the rate constants matrix contraction method developed by Sumiya and Maeda [101]. The energetic span model by Kozuch and Shaik uses the steady-state approximation to simplify the situation. A recent extension (gTOFfee) by Garay-Ruiz and Bo enables the energetic span to be calculated for a reaction network [102].

An illustration of the various methods to translate a reaction with two interconnected cycles (commonly occurring for asymmetric organocatalysis) with the work of Wong and co-workers is given in Figure 7 [103]. The authors used Kozuch and Shaik’s energetic span model (ESM) to calculate the TOF separately for the major *R*- and *S*-pathway and proposed, based on the TOF, that the *S*-pathway is favored. We note here that the model by Kozuch and Shaik does not consider coupled reaction pathways, as depicted in Figure 7. Both pathways share a common intermediate; thus, they will complete with the shared pool of CAT-HSPh. The impact of each competing elementary step captured by solving the system of ordinary coupled differential equations (ODE) numerically is similar to that derived by calculating the TOF for each pathway separately.

Translating computed quantities from the potential energy surface to experimental observables can be challenging and may lead to erroneous conclusions. This is especially important in cases where the solvent molecules are considered part of the reactive species and in PTC where insoluble solids are involved. Some examples will be highlighted below.

In the neutral hydrolysis of ester, Wang and Cao proposed the involvement of six water molecules (excluding the reactive water, a total of seven altogether) to lower the barrier of methyl formate hydrolysis [106]. Pliego and co-workers pointed out that the barrier is computed with the RC as the reference (a rigid cluster of methyl formate and seven water molecules, see Figure 5 for an explanation of RC) [99]. This neglected the penalty incurred from preorganization (Figure 5). However, in bulk water, water molecules are expected to be present as hydrogen-bonded oligomers [107]; therefore, taking the reference point as seven isolated water monomers, as proposed by Pliego and co-workers, may not be the most accurate representation. More discussion on this point will be presented in the “2.7.1. Examples Related to Solvation” Section.

One of the challenges in computing a realistic reaction profile involves obtaining the free energies of the reactants (such as G_A_ and G_B_ in Figure 5). The starting points (reactants, reagents, and catalysts) are the reference from which the relative free energies of all critical intermediates and TS are calculated. Thus, it is essential to compute key metrics such as TOF or experimental rate constants.

In PTC, inorganic salts (fluoride, carbonates, hydroxides of group 1 and 2 metals) are frequently employed. The energy calculated for an isolated molecule of these inorganic salts is unlikely to be representative of the bulk. As such, some groups have reported a hybrid approach where experimental data for these inorganic salts are combined with calculations in a thermodynamic cycle. 

Pliego and Riveros corrected their crown ether-catalyzed nucleophilic substitution of alkyl bromide with KF with experimental standard free energies of formation for KF and KBr [108]. In their reaction profile, the initial steps are omitted (complexation of KF to 18-crown-6); instead, the reaction profile begins with the exchange of the crown ether complex of KBr (18-C-6-KBr) with 18-C-6-KF (Figure 8). 

Another example by Carvalho and Pliego Jr is the nucleophilic fluorination under PTC with crown ether, where they considered the solvation of KF and KBr in toluene [109]. Gouverneur, Paton, and co-workers applied the same concept to obtain a more realistic reaction profile in their asymmetric nucleophilic fluorination under PTC. However, CsF and CsBr are used instead of KF and KBr [29].

Theoretical calculations of solid-state-related quantities will add another dimension of challenge to modeling PTC via computational chemistry. Using such a thermodynamic cycle is practical when the relevant experimental data is available. However, for many salts in PTC, such information is often absent.

### 2.6. The Computational Challenge—Mechanism

Prediction of experimental performance metrics such as TOF or selectivity for a complicated reaction is immensely challenging. Both the significant productive (leading to desired product) and non-productive (leading to side-products) pathways must be included. Hints from experiments (product distribution, kinetics, and spectroscopy) are frequently employed to reduce the number of possibilities and render computation more tractable. 

Nevertheless, strictly in silico catalyst design would require no input from experiments except for the final testing of the de novo designed catalyst. Algorithms are available to explore various reaction pathways automatically [110,111,112,113,114]. However, their widespread application in organocatalysis and PTC is still impeded by the availability of sufficiently accurate methods to calculate reaction barriers for realistic systems within a practical timeframe (see Table 3 for an estimate of the relative computational time for selected DFA), unless, of course, one has access to a tremendous amount of computing resources.

For asymmetric organocatalysis or PTC, the species involved in the critical enantioselectivity determining step must be known. Before any prediction of the level of enantioselectivity, the elementary step in which the chiral centers are formed must be known. In some cases, this might not be obvious. For instance, enantioselectivity is often improved using additives whose role is not immediately apparent. Nonetheless, it affects the level of enantioselectivity strongly suggests its involvement in the stereo-determining TS structures.

In PTC, the counter-cation of the bases or reagents, in some cases, has a profound influence on the reactivity, enantioselectivity, or both. For instance, in the *bis*-guanidium (BG) catalyzed enantioselective oxidation of sulfide, the counter-cation (Na, K, NH_4,_ or Ag) can have a profound effect on the sulfoxide yield and *ee* (Figure 9) [115]. This strongly suggests the involvement of the cation in the stereo-center forming TS structures. The loading of the NaH_2_PO_4_ was found to affect both the yield and *ee*. The authors proposed that two H_2_PO_4_ anions coordinate to a peroxo-tungstate species. This is supported by comparing the computed key vibration modes of the purposed species with the measured Raman spectrum. The roles of the cations in the critical TS structures and their interplay with H_2_PO_4_ remain to be elucidated.

Gouverneur, Claridge, and co-workers reported a detailed nuclear magnetic resonance (NMR) study on fluoride binding to their BINAM-derived *bis*-urea catalysts [116]. In their work, they unravel the interaction between fluoride and the catalyst. Under conditions closer to the reaction setting (CsF in CD_2_Cl_2_), they proposed two fluoride complexes mainly from ^19^F chemical shifts, T_1_ relaxation, and proton diffusion experiments. The major species, which is a 1:1 fluoride *bis*-urea complex ([*bis*-ureaCsF]), is proposed to be in rapid equilibrium relative to the NMR time scale with minor 1:2 fluoride *bis*-urea complex ([(*bis*-urea)_2_CsF]). The mixture of these two species at equilibrium proved competent in producing the same enantioenriched product when the substrate is added. The molecular structures and the role these two species may play in the reaction provide further opportunities for computational work.

In another highly detailed mechanistic study from the same group, Wong et al. reported their work on azidation via hydrogen bonding PTC (Figure 10B) [117]. Kinetic modeling with in situ infrared measurement of the product concentration was performed. Two parallel pathways were proposed based on the measured data—a racemic uncatalyzed pathway and an enantioselective catalyzed one. Relevant constants related to the rate of these two pathways were obtained. The catalytic cycle involves the exchange of ions between different species. Solid NaCl was proposed to be an intricate part of the catalytic cycle, where it plays an inhibitory role. DFT calculations, together with extensive conformational sampling with *bis*-urea and azide, gave a calculated % *ee* of 32, which is in qualitative agreement with the experimental value (62% *ee*).

Three potential complications can arise in studying PES via the location of key stationary points and subsequently deriving related kinetic properties from it.
Active species involved in catalysis: Aggregation of catalysts is well-documented, and the non-linear effect is closely associated with such a phenomenon.Validity of the transition state theory:Solvation: Extensive discussion on the choice of theoretical methods based on high-quality benchmarks is restricted to gas-phase modeling. Organocatalysis and PTC inevitably are solution-based chemistry. Therefore, the influence of solvents, which can be game-changing at times, often needs to be addressed.

Further elaboration on these three points will be provided in the following subsections.

#### 2.6.1. Non-Linear Effect in Asymmetric Catalysis

A non-linear relationship between the catalyst % *ee* and the product % *ee*, known as the non-linear effect (NLE) [118,119], signals the aggregation of catalyst molecules. However, one should note that the observation of NLE does not guarantee the involvement of catalyst aggregation [120,121]. Various classic examples are given in the review of Kagan [122]. Some recent examples in organocatalysis where NLE is observed will be discussed below.

Zhao and co-workers reported the observation of a negative NLE in their difunctionalization of allylic sulfonamides via iodination [123]. Together with observing a fractional reaction order with respect to the catalyst, the authors proposed the involvement of catalyst aggregation in the resting state (Figure 11A). However, the exact size of the aggregate is not known. Sun, Huang, and co-workers observed a positive NLE in their chiral phosphoric acid-catalyzed synthesis of α-amino esters [124]. They attributed the NLE to the catalyst’s heterochiral aggregates, which have low solubility at the reaction temperature. Jørgensen and co-workers reported a positive NLE in a proline-catalyzed [12 + 2] cycloaddition [125]. They proposed that two proline molecules are involved in the stepwise conjugate addition-carbonyl addition sequence (Figure 11B). DFT calculations supported their proposal. However, Burés, Blackmond, and co-workers performed kinetic simulation on an alternative mechanistic scenario. They concluded that the observed NLE in this [10 + 2] stepwise cyclization could occur due to a complex network of reversible steps instead of involving more than one catalyst molecule [126].

In general, the aggregation of catalyst molecules could present a significant challenge to calculating key reaction performance metrics such as TOF or *ee*/*er* from investigating a PES for the following reasons. (1) Solubility of the aggregate generally cannot be readily obtained from the study of PES. (2) The degree of aggregate is usually not known. (3) Aggregation would render the location of TS structures or equilibrium geometries computationally impractical for large catalyst molecules. However, the calculated energy profile of a proposed mechanistic network coupled with kinetic modeling can determine the extent of NLE and, when compared with experimentally measured NLE, provide strong evidence to corroborate the proposed mechanism.

#### 2.6.2. Bifurcating Potential Energy Surface

The TST is the cornerstone to in silico prediction of TOF and stereoselectivity, as discussed in the preceding relevant sections. The TS structure or an index-one saddle point has one imaginary frequency. The intrinsic reaction coordinate (IRC) is used to follow this imaginary frequency to determine the two equilibrium geometries (commonly referred to as the reactant/pre-TS complex and the product/post-TS complex) joined by the obtained TS structure. It is assumed that one TS structure will lead to one product via the minimum energy path (MEP) obtained from an IRC calculation. However, it has been extensively documented that many organic reactions exhibit post-TS bifurcation [127,128,129,130]. The TS structure exhibiting such behavior is called an ambimodal TS structure. Bifurcation at the “valley-ridge inflection” point allows more than one product to be accessed via one TS structure. An example of such a PES, generated from the model potential reported by Collins et al. [131], is given in Figure 12.

One of the key assumptions in the TST is that the redistribution of excess internal energy amongst the vibrations and internal rotation is much faster than a reactive elementary step—an assumption that may not always hold. In post-TS bifurcation, a valley-ridge point exists as one traverses the energetic landscape from the reactive complex through the TS to the product.

An example relevant to organocatalysis and PTC would be the S_N_2 of alkyl fluoride (Figure 13) [132,133]. In the gas phase S_N_2 of thiolate with methyl halide reported by Longo and co-workers [133], with methyl fluoride as the electrophile, the predominant trajectory is *tr*A which corresponds to a bifurcation following the IRC path (*tr*C). For methyl bromide, most of the trajectory bypasses the IRC path to form the S_N_2 product direct (trN). The resulting product is HF and methyl thiolate.

Prediction of the level of enantioselectivity in asymmetric organocatalysis or PTC inevitably relies on the use of the TST. The relative energies of the ensemble of TS determine the enantioselectivity. To our knowledge, assessing the importance of bifurcating PES in organocatalysis and PTC has not been reported. Determining the level of enantioselectivity is one potential opportunity to explore post-TS bifurcation further, as the Curtin–Hammett scenario, which is central to selectivity prediction, is rendered invalid—the relative free energies of all relevance ensemble of TS structures cannot be used to predict stereoselectivity or product distribution. A recent example by Singleton, Larionov, and co-workers on the stereochemistry of a photochemically induced ring contraction is illustrated in Figure 14 [134]. While no catalyst is involved in this reaction, it is an example in which the stereochemistry cannot be determined by the location of two sets of diastereomeric TS structures—the TS bifurcates to form both R and S chiral centers!

Technically, running many MD simulations to obtain a statistical distribution of potential products from a bifurcating PES is very computationally expensive. While, given the capability of modern hardware and software, it is not an impossible task, an approximate method to obtain the same outcome with less computation cost is nonetheless highly desirable. To this end, Lee and Goodman ValleyRidge PYTHON script can be used [135]. Srnec and co-workers presented the reactive mode compositive vector analysis to determine the ratio of products that arise from the bifurcation [136]. Other methods to approximate the bifurcation product distribution include the work from the groups of Carpenter [137], Houk [138], and Schmittel [139]. The work by Gomez-Bombarelli and workers utilized a neural network potential to perform reactive MD and may potentially be applied to the study of bifurcation with reduced computational cost [140]. These tools are expected to play an important role in calculating the product distribution where post-TS bifurcation is important and can provide a preliminary indication of such behavior before one decides to invest in a more expensive MD simulation.

### 2.7. Modeling Solvation

A solvent scan is one of the routines in developing an organocatalyzed or a PTC reaction, as the solvent can profoundly affect both the reaction yield and selectivity. Three possible approaches to model solvation are depicted in Figure 15. The importance of solvation in modeling and its recent advances and present limitation have been highlighted in several reviews [141,142,143,144].

The implicit solvation model is the most common and computationally economical method to account for some of the influence of the solvent. The polarizable continuum model (PCM) is the most common and accessible method to incorporate some solvent effects. In the PCM, the solvent influence is modeled as an ensemble average at thermal equilibrium, interacting with an electric “reaction field” [146]. The surface of the solute interacting with this field can be constructed by rolling a probe of fixed radius over the solute to generate a solvent-accessible surface (SAS). For the caveats and considerations when using a PCM in the study of key points on the PES, the readers are referred to the work of Coote and co-workers [77], and Truhlar, Cramer, and co-workers [147]. The deficiency of the continuum solvation model for charged species has been extensively reported. In the current context, this will be important when overall charged species are considered (e.g., urea-fluoride complex).

The popular SMD solvation model of Truhlar and co-workers [148] is parametrized against experimental solvation data. Its usage on more exotic structures beyond its training set, such as those found in TS structures of organocatalytic or PT-catalyzed reaction, may pose a problem [149,150]. Nevertheless, many computational studies on organocatalysis utilizing this solvation model have successfully reproduced key experimental results (refer to Figure 2 and Section 2.8 Validation of Calculations).

On the other end of the spectrum is the explicit solvation model, where all solvents are included as part of the simulation. It is generally only physically meaningful when combined with calculations supporting periodic boundary conditions due to the large number of molecules required to model a condensed bulk phase—having many more molecules implies that the conformation degree of freedom would pose an often-insurmountable computational challenge to obtain key stationary points and subsequent calculation of thermochemical correction via statistical thermodynamics, as much it is common to use molecular dynamics simulation when explicit molecules of solvent are involved. However, elementary steps corresponding to most chemical reactions are considered rare events on the MD time scale. As such, enhanced sampling algorithms have to be utilized to bias the simulation and enable free energy to be calculated [151].

In between the two extremes is a hybrid approach where a few solvent molecules could be added to the key region or sufficient solvent molecules are added to form a solvation shell (micro-solvation). A PCM-based solvent model can then be applied to model the bulk solvation. The continuum-cluster approach involves adding a cluster of solvent molecules and using the PCM model together. The thermodynamic cycle has been used extensively to calculate the solvation-free energy of ions in water. Two approaches (Figure 16) exist to construct the thermodynamic cycle—monomer-based [152,153] or cluster-based [154]. They differ by whether individual molecules of solvent or a cluster of solvent molecules are considered. The work of Goddard and co-workers provides a detailed discussion of the various consideration (especially on the choice of standard state and the associated correction) when using such a hybrid approach [155]. The thermodynamic cycle concept was extended to include the solvation energy of the TS structure (which replaces ion(H_2_O)_n_ in Figure 16) in a chemical reaction, as exemplified by the work of Ho and co-workers. However, their accuracy (on ionic S_N_2 reactions) was found to be inferior to the free energy of reaction directly obtained from umbrella sampling calculations (vide infra) [55].

Constructing a converged micro-solvation shell(s) in a continuum-cluster approach is challenging. The quantum cluster growth (QCG) method of Grimme and co-workers handles this task in an automated fashion [157]. ABcluster [158] by Zhang and Dolg used a bee colony algorithm (for the location of global minimum). Basdogan and Keith have used it in their computational study of the Morita–Baylis–Hillman (MBH) reaction (vide infra) [159].

#### 2.7.1. Examples Related to Solvation

The number of atoms required in explicit solvation to produce experimentally realistic concentration is generally incommensurable with reasonable computational cost. Considering the typical concentration of an organocatalytic reaction, a concentration of 0.01 M is a reasonable estimate for the catalyst. At the atomic level, this would require a 5.5 nm cube which can contain tens of thousands of atoms (Table 4). Such a simulation is probably in the realm of linear-scaling DFT coupled with massively parallel computing resources. Another challenge will be that chemical transformation is generally considered a rare event in a molecular dynamics simulation. An enhanced sampling algorithm would need to be applied for them to be observed in a practical duration and obtain critical quantities such as free energy. Despite the increased probability of observing rare events, the number of MD steps to converge such a simulation is often prohibitively high for a DFA-based calculation suitable for modeling chemical reactions (e.g., hybrid GGA).

An uncatalyzed reaction at a high concentration (>1 M) will be more amenable to an explicit solvation study of its reaction. Gunaydin and Houk reported using metadynamics simulation to investigate the hydrolysis of methyl formate [160]. The MD simulation with BLYP DFA was performed in a 10.1 Å cube with one methyl formate and 30 water molecules ([methyl formate] ≈ 1.6 M). In their simulation, the hydrolysis proceeds via an auto-dissociation of water. The resulting proton activates the ester by coordinating with its carbonyl oxygen. The hydroxide attacks the carbonyl carbon to form a diol intermediate. The overall ΔG^‡^ to include the diol intermediate was reported to be 22.8 kcal/mol. We note that in a more dilute setting, the probability of a water molecule auto-dissociating near the ester would be drastically reduced. 

In the work of Pliego and co-workers discussed earlier on the hybrid approach by Wang and Cao to model ester hydrolysis [106], they argued that the reference to calculate the barrier should be taken as one methyl formate and seven isolated molecules of water (Figure 17) [99]. In their calculations, this translates to a barrier of 77.2 kcal/mol (or 60.5 kcal/mol after a correction with water concentration by RTln(55.4)). The main contribution to this large barrier was mainly from the loss of translational entropy when eight isolated molecules became one RC. The translational entropy contribution is converted to the liberation free energy in solution. It is thus unlikely to be as punishing as the authors’ calculation suggested [87]. In addition, Goddard and co-workers pointed out that the formation of a water cluster can be viewed as a rearrangement of liquid water into solvated water clusters with no associated change in free energy (ΔG*_aq, cluster_ in Figure 17) [155]. However, we note that the extent to which this holds for an aqueous solution of methyl formate depends on the solute’s concentration. Preliminary calculations in Figure 17 give a ΔG^‡^ of 32.5 kcal/mol with an aqueous cluster of 7 water molecules as a reference. While this is lower than the seven isolated water molecules case, it is still significantly higher than the reported barrier of 25.9 kcal/mol.

One avenue to accelerate the solvation model that includes explicit solvent molecules is the QM/MM method. The solvent molecules are described with a molecular mechanics (MM) method. At the same time, the catalysts and reactants are modeled at the DFT level. This can significantly accelerate the study of organic reactions with explicit solvation. This technique has been used for some S_N_2 reactions with ionic nucleophiles to compare various solvent models’ performance [55,161]. Ho and co-workers found that umbrella sampling combined with the ωB97M-V/ma-def2-SVP level of theory as the QM level provides superior results relative to thermodynamic cycle-based calculations when validated against experimentally reported S_N_2 reaction in various organic solvents [55]. Keith and co-workers reported a study of NaBH_4_ reduction of CO_2_ in water with various solvation models, including the more sophisticated form of PCM. They found that the PBE DFA overestimates the solvation energies when the ion pairs are in proximity. In conclusion, they recommended, based on their result, a hybrid approach or the use of a conductor-like screening model for realistic solvation(COSMO-RS) [162] and reference interaction site method (ESM-RISM) [163] to account for specific interactions in ionic species [164].

In the context of organocatalysis, the Morita–Baylis–Hillman reaction, with its zwitterionic species and strong solvent dependence, remains challenging to model. Following the detailed experimental measurement reported by Plata and Singleton [82], ΔG*^‡^ and ΔG* reactions for several elementary steps were obtained. Sunoj and co-workers attempted to reconcile the difference between values calculated from key stationary points and the experimentally reported values [86]. They used DLPNO-CCSD(T) values to minimize gas phase electronic energy errors. Solvation energies of key stationary points calculated by the free energy perturbation (FEP) method with explicit solvent molecules (modeled with classical forcefield) provide an improvement over the implicit solvation model in some cases, but stepwise proton transfer to form the enolate remains in disagreement (Figure 18). We emphasize that due to the experimental limitation (e.g., unable to observe the change in concentration of all proposed intermediates under realistic experimental conditions), some assumptions need to be made in calculating and assigning values to each point in the free energy profile. Plata and Singleton pointed this out in their work [82]. Basdogan and Keith, in an extensive study with a diverse set of computational techniques for the same reaction, found that with five explicit MeOH molecules, they obtained the best agreement with experimental values [159].

The inclusion of solvent effects is challenging in PTC, as the key elementary steps could occur at the interface where the solvents’ properties differ significantly relative to their respective bulk properties. For instance, solvents with very similar dielectric constants (the basic parameters in the PCM solvation model) can vary substantially in % *ee* in asymmetric PTC. The work of Tan and co-workers on the pentanidium-catalyzed conjugate addition reveals such a phenomenon (Figure 19) [145]. The product yield and % *ee* decreased when the solvent was switched from toluene to *o*-xylene. Both solvents are aromatic hydrocarbons that differ only by a methyl group in the ortho position. Single-point calculations with the SMD solvation model indicated no significant difference in the relative energy of the two key TS structures (as reported by us [165]) in toluene and *o*-xylene. Chin and Krenske highlighted the limitation of the implicit solvation model in describing ion pairs in their computational study of CPA-catalyzed asymmetric Nazarov cyclization, which could be significant in the case presented here [76].

β-cyclodextrin(CD) is an inverse PTC, as it transfers organic molecules into the aqueous layer. Elk and Benjamin studied the phase transfer process of β-CD in a biphasic layer of water and 1-bromobutane [166] or 1-bromooctane [167]. Classical forcefields were employed in these studies. The same group also reported an empirical valence bond (EVB) approach to study the β-CD catalyzed S_N_2 reaction of CH_3_Cl and chloride via umbrella sampling with explicit solvation [168]. Their calculations indicate that the reaction barrier with β-CD complexation and in its absence is similar in bulk water. However, at the liquid/liquid interface, β-CD complexation reduces the reaction barrier by 2.4 or 3.4 kcal/mol relative to the uncatalyzed reaction.

### 2.8. Validation of Calculations

As in the case of machine learning, validation and testing lend credence to the predictive capability of computational methodologies employed in organocatalytic calculations. This is especially important due to the many layers of approximations and assumptions involved in arriving at predicted values.

Validation with experimental TOF is more challenging as most organocatalysis or PTC works do not measure the TOF of their reported reaction but only the isolated yield at the end of a specific duration. Measurement of TOF experimentally is cumbersome, as several points must be sampled accurately and fitted to a function where the gradient at the desired point in time can be calculated. For such purposes, the isolated yield after a purification process (such as column chromatography) is unlikely to reflect the catalyst efficiency accurately due to losses. In situ spectroscopic measurement (to obtain the spectroscopic yield) is the preferred method for such a purpose. Calculation of the TOF of a product from key stationary points is highly challenging, as it requires the critical productive and non-productive pathways to be known. In addition, modeling the thermodynamic and kinetic of phase transfer processes in PTC, which is highly challenging to model accurately, will add to the challenge. Advances in automated reaction pathway exploration algorithms coupled with a fast and reliable means to evaluate the energy and forces of a molecular system will be imperative to systematically obtain all significant pathways which the reactants and catalysts can access under a given reaction condition. In silico prediction of TOF in organocatalysis and PTC remains a challenge to be surmounted.

Some examples of detailed kinetic studies in organocatalysis include guanidine and its derivatives [169,170], proline [171,172,173], and thiourea/urea [174]. More recent examples in organocatalysis or PTC include the work by Jacobsen and co-workers [88] and Gouverneur and co-workers [117]. However, performing a PES study to locate all the required stationary points required to reproduce the experimentally measured rate constants is going to be highly challenging. As such, they are not attempted even in these highly detailed studies. In addition, the intrinsic error associated with absolute rate measurement will impede its effectiveness in validating against theoretically derived ones.

#### 2.8.1. Validation: Heavy-Atom Kinetic Isotope Effect

When judiciously applied, the kinetic isotope effect (KIE) can provide insight into the nature of the TS [175]. Coupled with the calculated TS structure, KIE can provide a multi-point validation for theoretical calculations. The synergy between heavy atom KIE and theoretical modeling via the location of TS structures, together with their associated challenges, are presented in detail by Lloyd-Jones and co-workers [176]. Proline catalysis will be a classic example where ^13^C KIE was utilized in conjunction with the theoretical calculation [177]. Two selected recent examples in organocatalysis will be highlighted below.

Kwan, Jacobsen, and co-workers utilized a distortionless enhancement by polarization transfer (DEPT) technique to measure the ^13^C/^12^C KIE in a thiourea-catalyzed glycosylation (Figure 20). Conventionally, an S_N_1-based mechanism gives rise to an inverse or small normal KIE. In contrast, an S_N_2-based mechanism gives rise to a large normal KIE [178]. However, in thiourea-catalyzed glycosylation where an asynchronous S_N_2, which involves the formation of a loose ion pair by dissociation of the leaving group followed by a rate-determining nucleophilic addition, can occur [179]. In conjunction with TS structure calculations, the authors provide evidence for an asynchronous S_N_2 by comparing calculated S_N_2 KIE at C1 and S_N_1 equilibrium isotope effect (an approximation to S_N_1 KIE) at C1 with experimentally measured values.

Hirschi, Mukherjee, and co-workers used ^13^C KIE as evidence to differentiate between various mechanistic scenarios. Experimental and calculated ^13^C KIE synergistically provide support for an E1cB mechanism [181]. They observed a significant inverse ^13^C KIE, usually associated with increased steric congestion at the carbon center on C2 (Figure 21). Based on this, they eliminated the C-C bond formation as the first irreversible step. Instead, they proposed nitro group elimination as the candidate. The E1 and E2 pathways were eliminated by their insurmountable barriers under the reaction condition. The ^13^C KIE work further reinforced this point.

#### 2.8.2. Validation: Enantioselectivity

Asymmetric organocatalysis provides a testing ground for modern computational chemistry. A metric of enantioselectivities such as *ee* or *er* can be measured reasonably accurately if sufficient care is taken. Theoretical calculation via the stationary point approaches does not usually require a highly detailed map of the reaction energy profile but only the key TS structures of the stereocenter-determining steps. The Curtin–Hammet principle is then invoked to determine the level of enantioselectivity by using a Boltzmann distribution on the ensemble of TS structures [16,183].

Due to the non-linear relationship between % *ee* or *er* with ΔΔG^‡^, an error of 0.5 kcal/mol can translate to a very different magnitude of error in % *ee* (Figure 22). For instance, at 253.15K, 0 to 0.5 kcal/mol of ΔΔG^‡^ corresponds to 0 to 46% *ee*, but from 2.0 to 2.5 kcal/mol, the difference in % *ee* is only 2.3%. This implies that at high % *ee*, the room for error in prediction is more forgiving than at low % *ee*. Therefore, a computational study that successfully predicts high enantioselectivity is often insufficient to showcase its predictive capabilities. Based on this analysis, we will showcase some selected examples in organocatalysis that demonstrate such predictive capabilities. 

Ermanis, Phipps, Goodman, and co-workers amply demonstrated this in their work on the chiral phosphoric acid-catalyzed enantioselective Minisci reaction (Figure 23). They observed a positive NLE, which is attributed to the formation of the insoluble heterogeneous dimer that is not directly involved in the main productive pathway [31]. The authors’ calculations can reproduce the variation in experimental % *ee* with various substrates. In this case, no dispersion correction is applied in the geometry optimization. However, their choice of method is sufficiently accurate to reproduce key experimental trends in % *ee* with selected examples from (70% *ee* and beyond).

We have demonstrated in the bicyclic guanidine catalyzed reaction between *N*-phenylmalemide and anthracen-9(10H)-one that our calculations can reproduce the difference in % *ee* between the *tert*-butyl and phenyl bicyclic guanidine (Figure 24) [32]. Our calculated % *ee* indicates that 4,5-dichloroanthracen-9(10H)-one has a lower level of enantioselectivity than reported by the authors [185]. Upon closer inspection of their supporting information, the reported 99% *ee* is likely overestimated. This highlights the experimental errors associated with high % *ee*. The minor enantiomer peaks might have significant noise from the baseline if the sample used for analysis is too dilute or the integration of peaks is done incorrectly.

In their study of SPINOL-phosphoric acid (SPA) catalyzed, Wheelers and co-workers show that the improvement in the level of enantioselectivity by varying the SPA’s side arm can be reproduced by their modeling (Figure 25) [34]. While the difference in % *ee* between calculated and experimental can be 10% or more in magnitude, the reader should note that such differences, when translated to ΔΔG^‡^, can range from a difference of 0.14 kcal/mol to 1.1 kcal/mol due to the non-linear relationship between % *ee* and ΔΔG^‡^ (see Figure 22 and Figure 25). Since chemical accuracy is usually defined to be 1 kcal/mol or less, good agreement with the experimental % *ee* is generally observed (near the limit of chemical accuracy, refer to values in parentheses in Figure 25), ranging from the low % *ee* of the 2,6-dimethylphenyl group to the drastic improvement in phenyl and 9-anthracyl group.

Jacobsen and co-workers reported a detailed mechanistic study on the ring-opening of oxetane with TMSBr (Lewis acid mechanism) or HBr (Brønsted acid mechanism) catalyzed by chiral squaramide (Figure 26). The TS structures for the critical enantioselective steps were located at SMD(Et_2_O)/B97D/def2-SVP [88]. For a set of seven catalysts variation for the Lewis acid mechanism, they explored the use of two different quasi-RRHO corrections to low vibrational frequencies and RRHO, where they found that the use of quasi-RRHO correction is generally beneficial. Some of the enantioselectivity appears to arise mainly from vibrational entropy, as they are highly sensitive to the correction used (see also the work of Chin and Krenske [76]). The quasi-RRHO correction by Truhlar reproduces the low level of enantioselectivity for the 4-fluoro- and 3,5-difluoro-phenyl groups. The decrease in % *ee* from the 9-phenanthryl to 1-naphthyl catalyst cannot be reproduced by applying a quasi-rotor correction.

These selected, but not exhaustive, examples demonstrate the potential of DFT calculations for in silico design of asymmetric reaction. However, from a practical point of view, given current technological limitations and costs, one would still need access to a considerable amount of computing power to perform these calculations within a competitive timeframe with experimental screening. 

### 2.9. Understanding/Interpretation

The activation strain model (ASM) model decomposes the activation barrier into two contributions: (1) the energetic penalty imposed to distort the reactants and catalyst from their lowest energy conformers to adopt the reactive conformation and (2) the interaction between distorted components [187,188]. The second part requires an energy decomposition analysis (EDA) which will be described in more detail below. The open-source PyFrag 2019 code can be used to perform the ASM calculations. Readers are encouraged to refer to the work of Hamlin, Bickelhaupt, and co-workers for more detail [189]. ASM can be applied to asymmetric catalysis instead of a reaction pathway, where the difference in geometrical strain/distortion between the pair of stereo-differentiating TS structures can be compared.

In organocatalysis or PTC, energy decomposition analysis (EDA) allows the interaction energies between the catalyst and substrates in the crucial TS structures to be decomposed into different contributions (electrostatic, exchange-repulsion, and dispersion). Neese and co-workers developed the local energy decomposition (LED), which uses a super-molecular approach to decompose the DLPNO-CCSD(T) interaction energy [190]. The LED decomposes the interaction energy into electronic promotion, electrostatic, exchange, dynamic charge polarization, and dispersion contributions. An application of LED is discussed below.

List and co-workers’ silyated C-H acid catalyzed enantioselective Diels–Alder [191] was studied in detail by Bistoni and co-workers [40]. Both LED and ASM were applied in their work. Through LED of DLPNO-CCSD(T) interaction energy, the authors showed that dispersion interaction is the primary source of energetic difference between the two diastereomeric TS structures (Figure 27). ASM was used to quantify the geometry distortion between the two TS structures. 

The non-covalent interaction Index (NCI) is a convenient way to visualize the non-covalent interactions between fragments in a molecular system [192]. It is based on the dimensionless reduced density gradient (RDG). The sign and magnitude of the RDG can be used to classify the type of interaction ranging from strongly attractive to strongly repulsive. NCI isosurfaces can be generated with NCIplot [193] or Multiwfn [194]. It can be used in organocatalysis to illustrate the key noncovalent interactions in critical TS structures. We will demonstrate this with the work of Bistoni and co-workers, as discussed above. The NCI isosurfaces indicate the difference in *van der Waals* type of interaction between the TS that leads to the major product—the *S, S* enantiomer, and that which leads to the minor enantiomer. The interaction between the naphthyl group of the catalyst and the cyclopentadiene is missing (Figure 27), which is consistent with the LED result.

The independent gradient model (IGM) developed by Hénon and co-workers is based on electron density contragradience [195]. The NCI and IGM isosurfaces of C60@6-CPPA based on DFT densities are shown in Figure 28. Interestingly, despite the repulsive interaction between C60 and 6-CPPA at B3LYP/6-31G(d) calculated at +18 kcal/mol, NCI and IGM isosurfaces indicate that their interaction is weakly attractive.

### 2.10. Machine Learning and its Relevance to the Field 

Machine learning (ML) is a ubiquitous topic in contemporary research and is currently a very active area. It is beyond the scope of this review to provide details into the multitude of applications that ML has impacted chemistry. Our emphasis will be on aspects of ML that can be utilized in in silico prediction of catalysts’ performance (yield, TOF, and stereoselectivity). Therefore, unsupervised learning-based works will not be discussed, although they are of great importance [107,197,198,199]. Readers are encouraged to read authoritative reviews and perspectives on these topics and the references cited to better appreciate ML’s depth and impact in chemistry [200,201,202,203,204,205,206,207].

#### 2.10.1. Supervised Learning

As discussed in the preceding section, in silico calculation of reaction efficiency metrics such as TOF or TON is challenging. In supervised ML, reactants (including catalyst and solvent) can be transformed into features or descriptors which can be mapped to the reaction yield via a model. Doyle’s group used results from high-throughput experiments of C-N cross-coupling with ML to predict reaction yields [208]. The use of ML to predict yields or turnover numbers in synthetic chemistry has been reported for transition metal-catalyzed asymmetric reactions [209,210,211], Heck reaction [212], multi-component reactions [213], and Michael addition [214].

Pertinent to asymmetric organocatalysis, Denmark and co-workers developed the average steric occupancy (ASO) descriptor to predict the level of enantioselectivity [215,216]. In later work by the same group, the use of random numbers to replace the features has pointed to a potential problem of bias in the splitting of dataset training randomly [217]. This problem was reported earlier by Chuang and Keiser, who employed two classical tests of generalization to Doyle and co-workers’ data: random number as features (straw model) and one-hot vector, which encodes the presence or absence of reaction components [218].

The number of data points required to train a meaningful model is generally incommensurate with organocatalysis methodology development. For instance, 1000 data points would be considered a tremendous amount of experimental work. However, from an ML perspective, it may be wholly insufficient (depending on the diversity of the data points). A methodology-developing chemist (without access to a high-throughput facility) would be unlikely to generate this many data points throughout the optimization process. The diversity of the data points is crucial for ML. Skewed data points would result in the problems discussed previously. In Doyle and co-workers’ reply to Chuang and Keiser, they employed domain expertise (knowledge of the roles of reactants) to partition their reported dataset and showed that the one-hot vector performance is no longer comparable to the chemical-based features model [219].

Feature engineering is an intrinsic part of ML. In the quest to develop generally applicable and computationally economical descriptors for general modeling reaction yield and selectivity, Glorius and co-workers engineered features or descriptors from multiple molecular fingerprints [220]. Luo, Zhang, and co-workers used an ensemble of structural and physical organic chemistry (SPOC) descriptors [221].

The features employed in ML reaction studies are continuous or discrete numbers derived from either calculated properties or experimental observables. In the case of organocatalysis/PTC or general homogeneous catalysis, the catalysts have well-defined connectivity between their atoms that will be lost when translated into a scalar descriptor/feature. These features/descriptors cannot precisely be reversed and mapped to a catalyst’s molecular structure, impeding their use to design novel and more efficient catalysts. Ideally, one would use an optimization algorithm to search the features’ space for an improved catalyst version, but this is usually not possible if the features cannot be reverse-mapped to a valid experimental condition (including catalysts, additives, or reactants). The inverse design of reaction from a supervised learning model based on these features remains challenging.

#### 2.10.2. Machine Learning Potential

At the heart of an in silico calculation of catalytic performance is the capability to calculate the energy and force of a system with various workflows. The ability to compute these quantities accurately and in a timely fashion is imperative in driving in silico reaction/catalyst design. In a machine learning potential (MLP), atom-based descriptors are mapped to energy, forces, and stress tensors. The model can be kernel-based or neural network. It can reproduce the accuracy of the dataset it is trained in a fraction of the time. For a more in-depth appreciation of MLP, the reader is referred to the review by Behler [222], Muller, Tkatchenko, and co-workers [223], and Pinherio Jr et al. [224]. Nevertheless, the “no-free-lunch” principle applies; significant effort has to be invested in generating the dataset for training MLP, and the quality of MLP is limited by the dataset on which it is trained.

Several general-purpose MLPs suitable for organocatalysis have been reported in the literature, for instance, ANI-2x [225], DimeNet++ [226], GemNet [227], and SchNet [228]. The Open Catalyst project also provides state-of-the-art MLPs that can be used out-of-the-box, although these MLPs are targeted toward heterogeneous and electro-catalysis [18,19].

A limited example is provided in Figure 29. We evaluated two MLPs on the relative potential energies of three critical TS structures in our work on the guanidine-catalyzed cycloaddition of Anthrone [32]. The time required to calculate potential energy with these MLPs is insignificant relative to the double hybrid DFA reference. GemNet-T can predict TS_SS_2 as the lowest energy TS structure, while ANI-2x predicts that TS_RR should be the most stable TS structure. The difference of less than 2 kcal/mol is noteworthy as these MLPs have not been trained on these TS structures. We use ANI-2x to perform a constrained optimization of TS_SS_2. The RMSD between the PBEh-3c and ANI-2x optimized geometries is 0.41, which is similar to that reported by Bistoni and co-workers when going from a double-zeta to a triple-zeta basis set as discussed above [40].

Here, we list some examples that could be of relevance to PTC. MD simulations with MLP on condensed phases have provided molecular insights into phenomena such as the dynamics of proton transfer in NaOH [229], dissolving salts in water [230], permeation of water into nanocapillaries [231], and zeolite hydrolysis [232]. Studies on aqueous hydroxide, commonly used in PTC, can potentially be extended to investigating relevant phase-transfer catalyzed reactions.

With NN potential, MD simulation with tens of thousands of atoms is generally within reach via a decent Graphic Processing Unit (GPU) accelerator. With sufficient computation resources, GPU accelerated MD simulations with deepMD-kit [233] have allowed the study of ice nucleation for a 300,000 atoms system on a SCAN DFA trained MLP [234], thus providing a super slow motion movie of physical phenomena resolved at the level of atoms. 

Challenges remain on whether the GGA and meta-GGA DFA typically used to train MLP have the appropriate accuracy in modeling the key interactions in organocatalysis and PTC. The compilation of DFA in Table 1 shows that computationally economical GGA falls short of the standard chemical accuracy requirement. Nevertheless, as the benchmark is referenced against calculations in a vacuum, it remains to be seen if such errors will translate into the condensed or the interphase. Careful validation with experimental properties is imperative to ensure that the trained MLP is physically accurate enough for the production simulation. Encouraging results from the group of Keith indicate that with sufficiently solvated intermediates and TS structures, the difference between GGA DFA, hybrid DFA, and DLPNO-CCSD(T) is generally slight [159,235].

Despite the shortcoming of GGA and meta-GGA DFA, generating hundreds of thousands of high-quality potential energy and forces with hybrid GGA and beyond will be an astronomical task. While there have been reports of MLP built on CCSD(T) [236] or DMC data [237], these works represent the exception rather than the norm. 

Technical challenges in generating a dataset for the training of MLP include k-points and plane-wave cut-off convergence. The use of empirical dispersion, critical to modeling noncovalent interaction, requires careful evaluation of k-points convergence due to the long-range nature of these corrections. Training of MLP is generally done with a smaller cell, and the resulting MLP is applied to a much larger cell. Care has to be taken to ensure that the smaller cell is large enough to contain the phonons or collective vibrations.

The selection of a representative dataset to train an MLP is challenging. Typically, MLP is trained with data generated from an ab initio MD simulation. However, only a small percentage of data from the simulation is used. One possible way to reduce the number of expensive DFT calculations is to utilize active learning. Active learning determines on the fly if a reference(usually DFT) calculation should be performed and whether this point has to be added to the training set [238,239,240].

Despite the challenges that remain, these works are encouraging, as with more advances in MLP design and training, scientists can couple these ML to conformational search algorithms, reaction path discovery algorithms, and molecular dynamics, thus enabling chemists to simulate system size and timescale that are closer to experiments. To end our discussion, we referred the reader to the comprehensive roadmap on ML in various subfields of chemistry for more detail [241].

## 3. Conclusions and Outlook

As noted by various authors, computational chemistry has advanced over the years as applied computational chemists can scale larger systems with improved hardware and software. Algorithmic advances in linear approximation to the CCSD(T) have provided the computational chemist with an additional method besides DFT to approach the problem. Nevertheless, the performance of such approximation on larger systems might not be equivalent to the smaller system on which they are benchmarked. Despite the advances in gas phase calculations, organocatalysis and phase transfer catalysis inevitably occur in the condensed phase. Solvation remains a challenge, and this aspect has been extensively discussed in this review and the references cited. 

Regarding achieving a practical in silico catalyst design, while algorithms to achieve automated exploration or mapping of key points on the PES is available, these remain highly expensive in terms of computational effort, thus impeding their widespread uses. Achieving a DFT or even CCSD(T) level of accuracy with forcefield-like computational effort will be imperative in driving practical in silico catalyst design. Houk and Liu predicted that an accurate polarizable force field could be attained by 2040 [21], while Grimme and Schreiner predicted that low-cost DFT capable of scaling to 10000 atoms would be available by 2043 [20]. Thus, if their prediction comes to fruition, and together with continued advanced in hardware, computational chemistry could be an intricate part of the methodology-developing chemists. This review ends with a discussion on ML potential, as we expect that MLP could assume the role of accurate polarizable FF or low-cost DFT.

Besides the pragmatic aspect of in silico catalyst design, the capability to perform molecular dynamics simulation with the number of atoms that approach the macroscopic scale will allow chemists access to atomic motions resolved at femtosecond resolution. The capability to perform MD simulation at such a scale can potentially provide us with the capacity to model solvents explicitly. Thus, obtaining accurate free energy of solvation can be used to validate and develop a more advanced implicit solvation model, which can drive catalyst development. The challenge of bifurcation PES can also be addressed. Together, they will bring us towards understanding and practical in silico catalyst design.

## Figures and Tables

**Figure 1 molecules-28-01715-f001:**
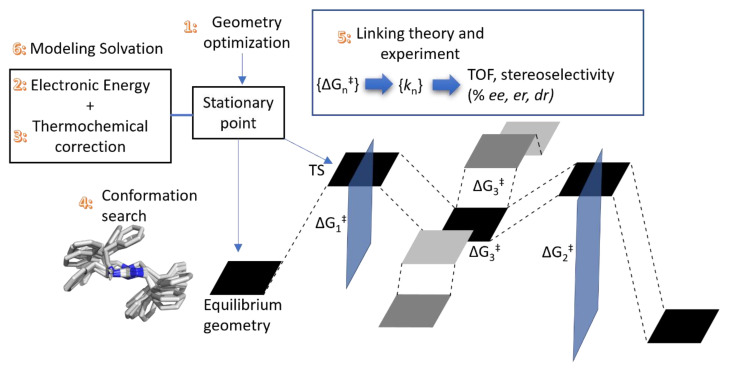
Summary of this review. Workflow to calculate a reaction performance metric from key stationary points on the potential energy surface. Numbers in orange corresponds to the sequence in which the mentioned topics will appear in this review.

**Figure 2 molecules-28-01715-f002:**
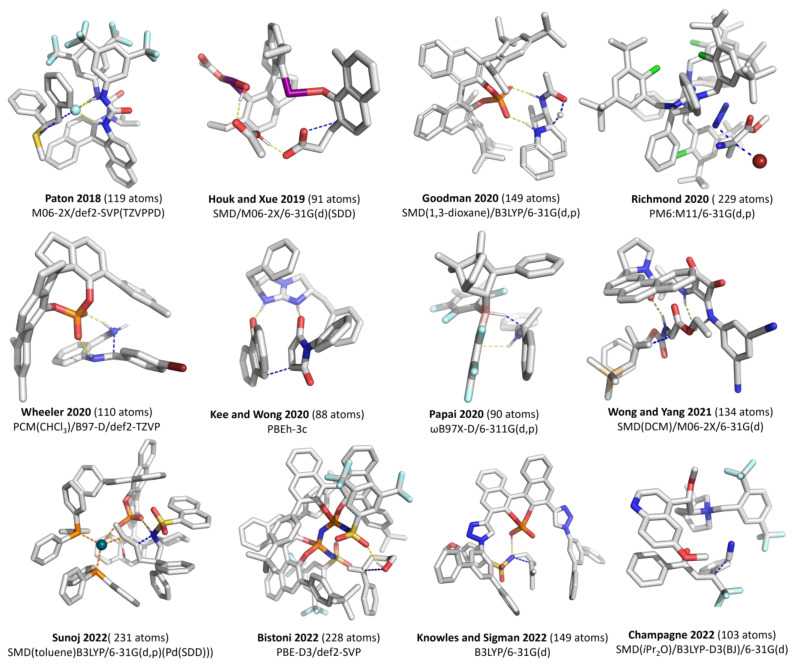
Selected examples of organocatalysts TS structures and their level of theory for geometry optimization. Yellow Dash line: hydrogen bond. Blue Dash line: key bonds in TS. Three-dimensional representations are generated from coordinates given in the cited references: Paton 2018 [29]; Houk and Xue 2019 [30]; Goodman 2020 [31]; Kee and Wong 2020 [32]; Richmond 2020 [33]; Wheeler 2020 [34]; Papai 2020 [35]; Wong and Yang 2021 [36]; Sunoj 2022 [28]; Bistoni 2022 [37]; Champagne 2022 [38]; Knowles and Sigman 2022 [39].

**Figure 3 molecules-28-01715-f003:**
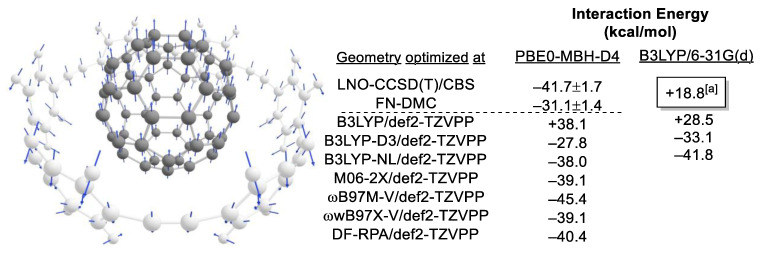
All values are in kcal/mol. LNO-CCSD(T) and FN-DMC values are taken from ref [71]. Values below the dotted line are calculated with ORCA 5.0.3 [66] using the following keywords: “DFA/def2-TZVPP autoaux verytightscf rijcosx” with the geometry supplied by ref [71]. DF-RPA is calculated with MRCC [67]. Blue arrows are the displacement vectors of the imaginary frequency at B3LYP/6-31G(d). ^[a]^ Interaction energy at the same level of theory as the geometry optimization, i.e., at B3LYP/6-31G(d).

**Figure 4 molecules-28-01715-f004:**
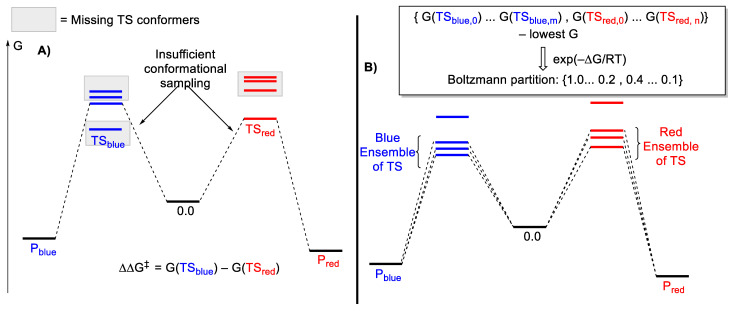
Hypothetical scenario (**A**) The lowest-lying TS_blue_ conformer is omitted. The opposite selectivity is thus predicted. (**B**) Many conformers with significant populations at the reaction temperature and pressure. An ensemble is required to obtain a more accurate calculated value of selectivity.

**Figure 5 molecules-28-01715-f005:**
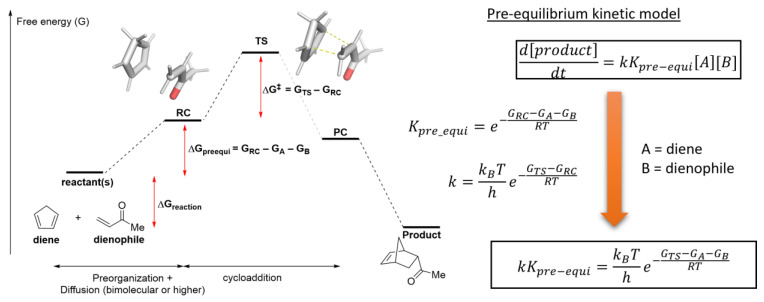
The reaction profile of a [4 + 2] cycloaddition. RC: Reactant Complex. PC: Product Complex. G is the free energy of the species labeled in the subscript. Reprinted with permission from ref [91]. Copyright 2016 American Chemical Society.

**Figure 6 molecules-28-01715-f006:**
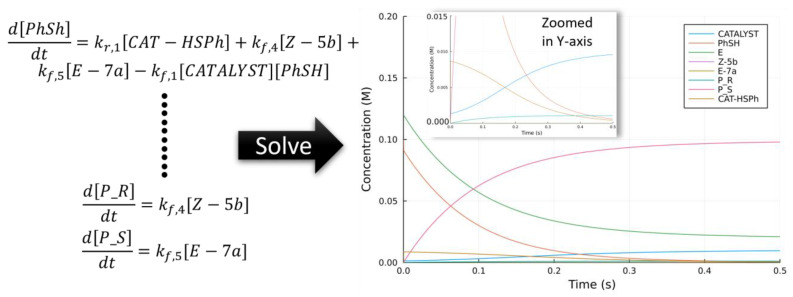
Solving a system of coupled ODE to obtain the concentration of all species over time (only selected equations are shown). The dotted line represented omitted equations.

**Figure 7 molecules-28-01715-f007:**
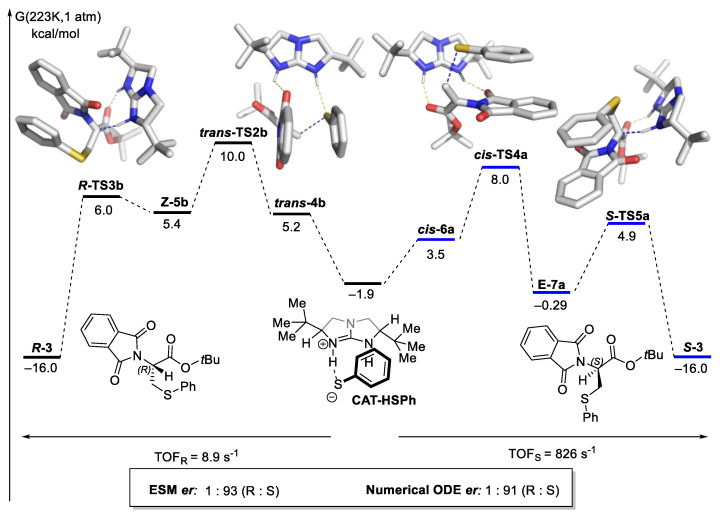
Reprinted with permission from ref [103]. Copyright 2012 American Chemical Society. TOF is calculated with AutoTOF from values supplied by the cited reference. Numerical ODE values are generated with DifferentialEquations.jl [104] and Arbnumerics modules in Julia [105]. The product formation is assumed to be irreversible. A pre-equilibrium kinetic model is used to obtain the rate constants from CAT−HSPh.

**Figure 8 molecules-28-01715-f008:**
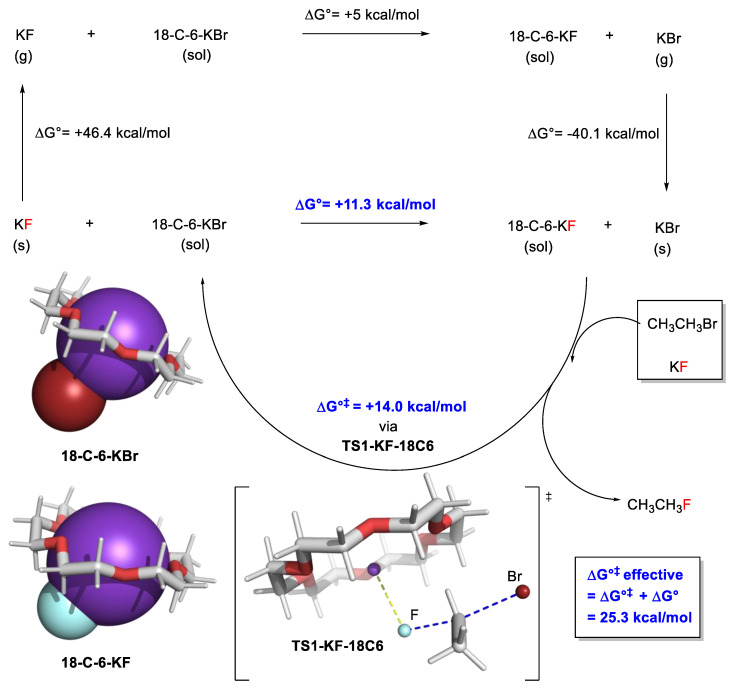
Use of thermodynamic cycle in 18-crown-6-catalyzed fluorination of alkyl bromide. All values are obtained from the work of Pliego Jr and Riveros in ref [108]. Fluoride is colored red to facilitate identification. (sol) refers to solution. (s) refers to solid.

**Figure 9 molecules-28-01715-f009:**
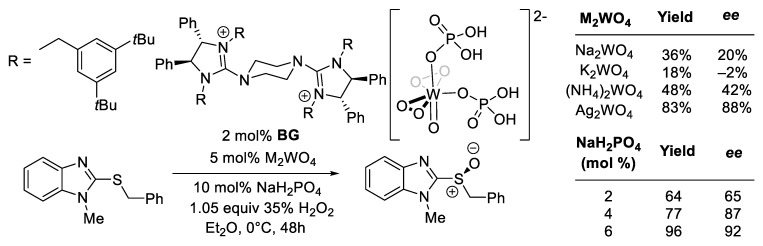
Enantioselective oxidation of sulfide reported by Tan and co-workers [115]. All values are obtained from the mentioned reference.

**Figure 10 molecules-28-01715-f010:**
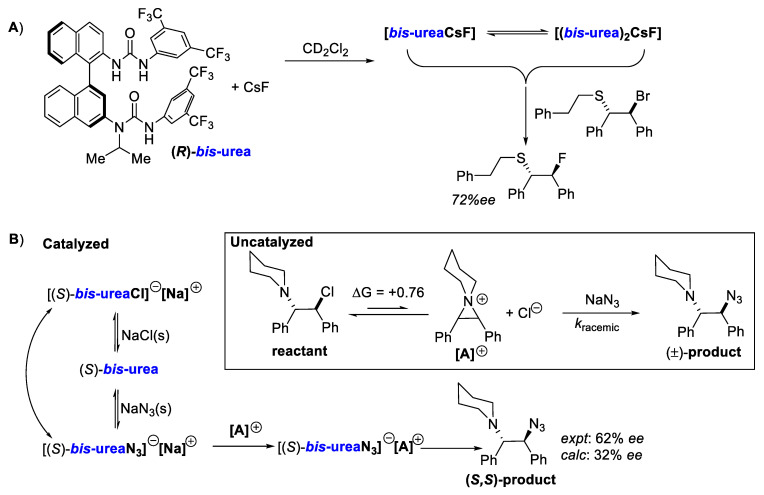
(**A**) Complexation of CsF and bis-urea catalyst as observed via NMR and proposed by Gouverneur, Claridge, and co-workers [116]. (**B**) Reaction pathways of *bis*-urea-catalyzed azidation were reported by Wong et al. [117].

**Figure 11 molecules-28-01715-f011:**
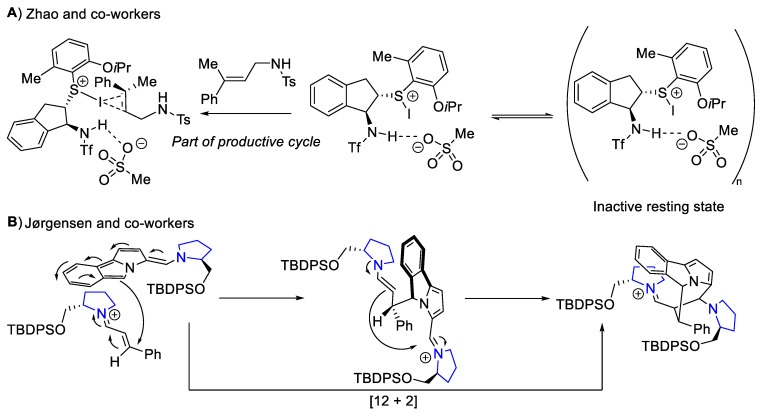
Examples of recent work with proposed NLE. (**A**) Difunctionalization of allylic sulfonamides via iodination as reported by Zhao and co-workers [123]. (**B**) Proline derivative-catalyzed [12+2] cycloaddition as reported by Jørgensen and co-workers [125].

**Figure 12 molecules-28-01715-f012:**
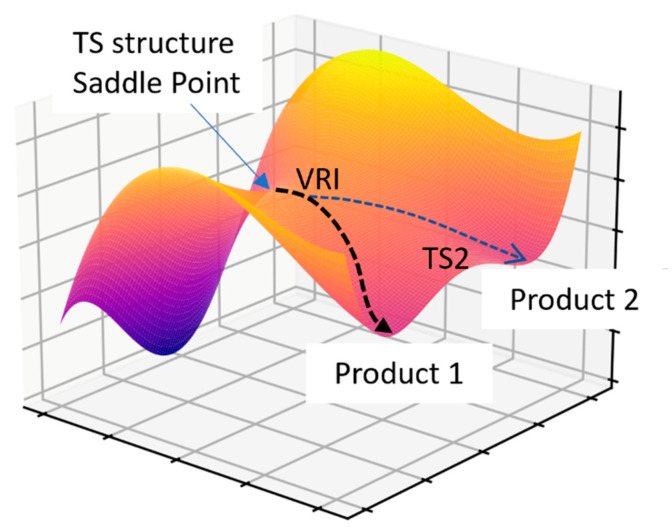
Bifurcation in PES. The PES is generated from the model potential provided by Collins et al. [131].

**Figure 13 molecules-28-01715-f013:**
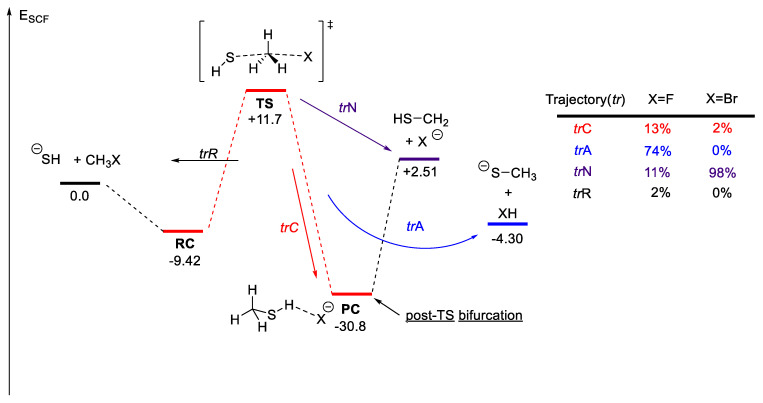
Values (in kcal/mol) calculated at MP2/6-311(d,p) are obtained from ref [133]. trC: approximately IRC path. trA: approximately IRC path followed by a proton transfer. trN: bypass the IRC path to form the product directly. trR: recrossing.

**Figure 14 molecules-28-01715-f014:**
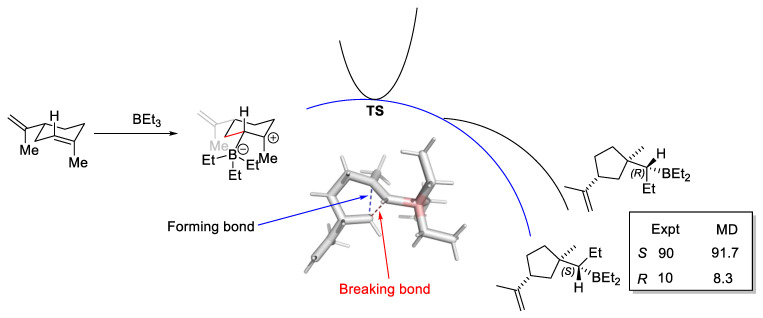
Singleton and co-workers reported an example of bifurcation leading to two stereochemical outcomes [134]. MD: selectivity predicted by a series of molecular dynamics simulations (all values are obtained from their work).

**Figure 15 molecules-28-01715-f015:**
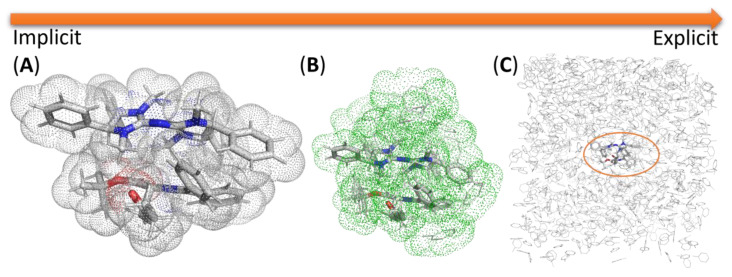
Various solvation models to apply to a PT catalyst that Tan and co-workers reported [145]. (**A**) Implicit solvation. Grey dots are SAS. (**B**) Hybrid approach. Solvent molecules (Toluenes) are displayed as wire. The green dots are the SAS. (**C**) Explicit solvation in a 5.6 nm cube. The catalyst is in the orange oval.

**Figure 16 molecules-28-01715-f016:**
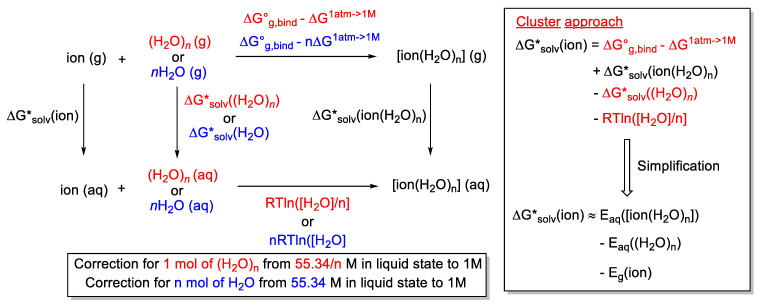
Thermodynamic cycle. * = standard state of 1 M, and ° = standard state of 1 atm. Red: cluster model. Blue: monomer model. ΔG^1 atm->1 M^ = RT ln(24.46) is the correction from 1 atm of an ideal gas to 1 M, where R = 0.08206 L atm K^−1^ mol^−1^ is the universal gas constant, and T is the temperature in Kelvin. E_aq_ is the electronic energy of the species calculated with implicit solvation, and E_g_ is the electronic energy in the gas phase. For assumptions involved in simplification, please refer to Wu and Kieffer [156].

**Figure 17 molecules-28-01715-f017:**
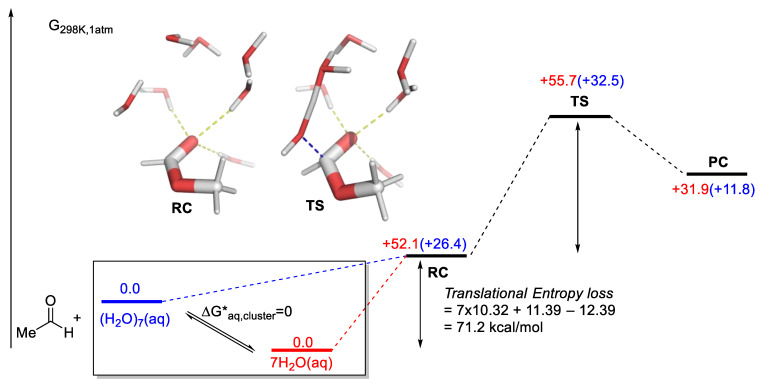
All values in the free energy profile are in kcal/mol and computed at CPCM(H_2_O at PBEh-3c)/DLPNO-CCSD(T)/CBS//CPCM(H_2_O)/PBEh-3c. CBS is based on a 2-point extrapolation with the def2-TZVPP and def2-QZVPP basis sets. For values in blue, the water cluster is the reference. For values in red, isolated water molecules are the reference. For comparison, key values from Pliego and co-workers [99] at MP4/TZVPP+diff//CPCM/X2LYP/6-31+G(d): ΔG^‡^(RC to TS) = 24.5 kcal/mol and ΔG(reference to RC) = +52.7 kcal/mol. For Wang and Cao [106] at CPCM/B3LYP/6-311+G(2df,2p): ΔG^‡^(RC to TS) = 25.4 kcal/mol. Correction of water concentration is not included.

**Figure 18 molecules-28-01715-f018:**
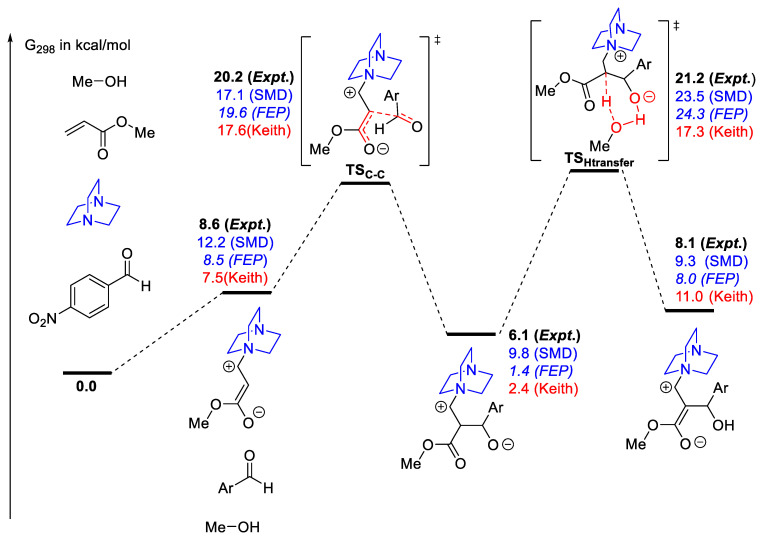
Partial free energy profile of MBH reaction. Experimental values are obtained from the work of Plata and Singleton [82]. Blue calculated values are from the work of Sunoj and co-workers [86] at DLPNO-CCSD(T)/CBS/B3LYP-D3/6-31+G(d,p) with solvation via SMD or FEP as indicated in parentheses. Red calculated values are from the work of Basdogan and Keith at DLPNO-CCSD(T)/def2-TZVP/BP86-D3/def2-SVP [159]. Explicit solvent molecules are not shown.

**Figure 19 molecules-28-01715-f019:**
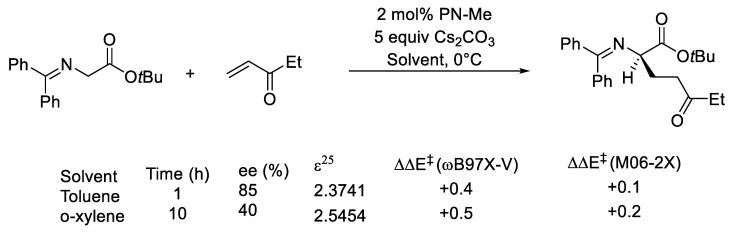
Pentanidium-catalyzed asymmetric conjugate addition. Experimental values are from the work of Tan and his co-workers [145]. ωB97X-V/def2-TZVPP and M06-2X/6-31+G(d,p) values included SMD solvation calculated with structures from Kee and Wong [165].

**Figure 20 molecules-28-01715-f020:**
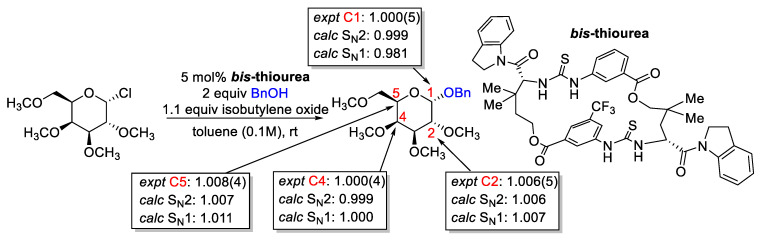
*bis*-Thiourea catalyzed glycosylation. All values are obtained from the work of Kwan, Jacobsen, and co-workers [178]. “*Expt*” refers to experimental ^13^C KIE relative to the C2 methoxyl group. “*Calc S_N_2*” refers to the calculated KIE value at PCM/PBE0-D3(BJ)/6-31G(d) and the Biegeleisen–Mayer method [180] adjusted with the Bell infinite parabola tunneling correction. “*Calc* S_N_1” refers to the equilibrium isotope effect.

**Figure 21 molecules-28-01715-f021:**
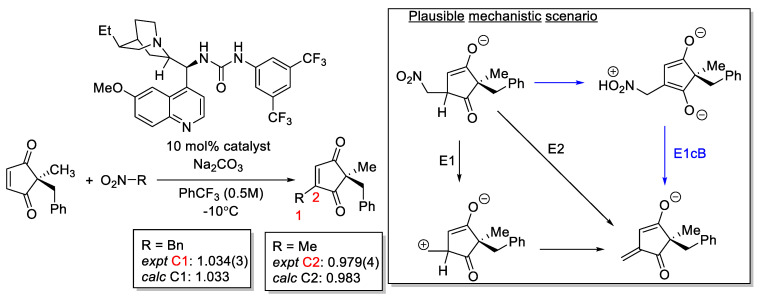
Urea-catalyzed desymmetrization [182]. All values are obtained from the work of Hirschi, Mukherjee, and co-workers [181]. “*Expt*” refers to the experimental ^13^C KIE relative (please refer to the cited reference for details. *Calc*” refers to the calculated ^13^C KIE at IEFPCM/B3LYP/def2-SVP and the Biegeleisen–Mayer method [180].

**Figure 22 molecules-28-01715-f022:**
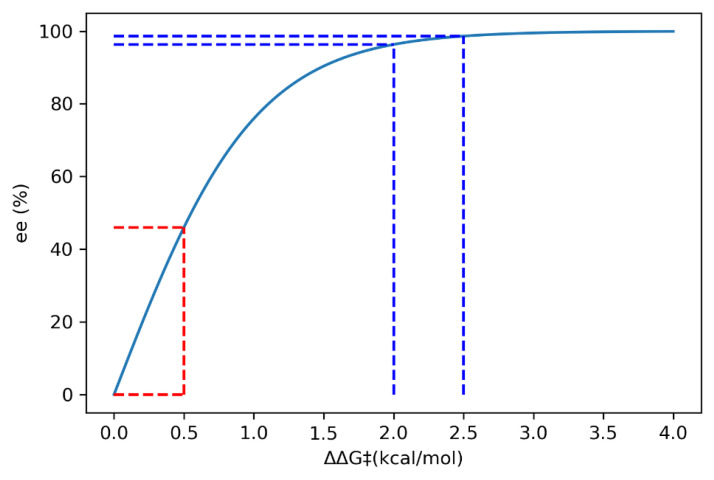
% *ee* vs. ΔΔG‡(kcal/mol) at 253 K. Two intervals of 0.5 kcal/mol are included to illustrate the difference in calculated % *ee*.

**Figure 23 molecules-28-01715-f023:**
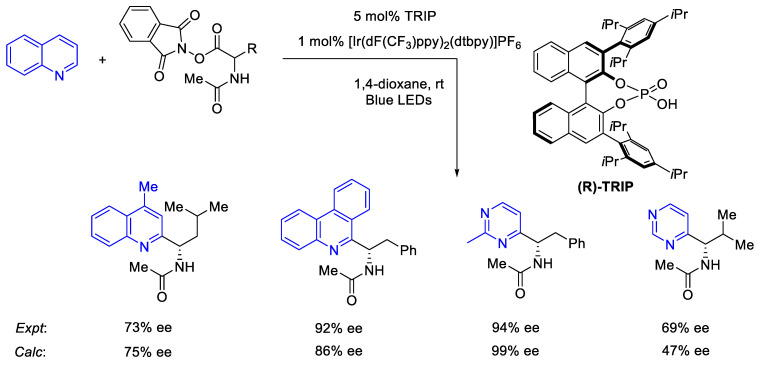
Enantioselective chiral CPA catalyzed Minisci reaction [184]. All values are from the work of Goodman and co-workers [31]. % *ee* are calculated from ΔΔG^‡^_298.15 K, 1 atm_ at SMD(1,4-dioxane)/M06-2X/def2-TZVPD//B3LYP/6-31G(d,p) from the cited reference.

**Figure 24 molecules-28-01715-f024:**
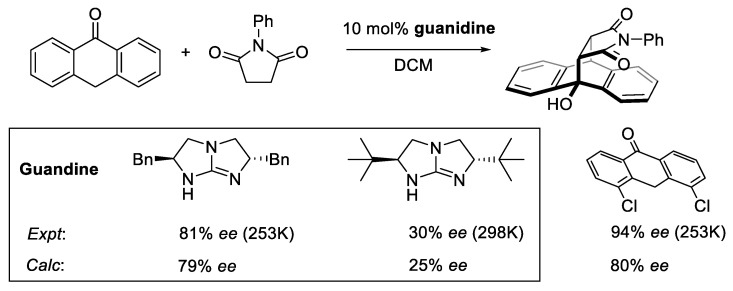
Bicyclic guanidine catalyzed cycloaddition [185]. Values are from the work of Kee and Wong [32]. % *ee* are calculated from ΔΔG^‡^ at CPCM(DCM)/DSD-PBE95-D3(BJ)/def2-QZVP//PBEh-3c as reported by the cited reference.

**Figure 25 molecules-28-01715-f025:**
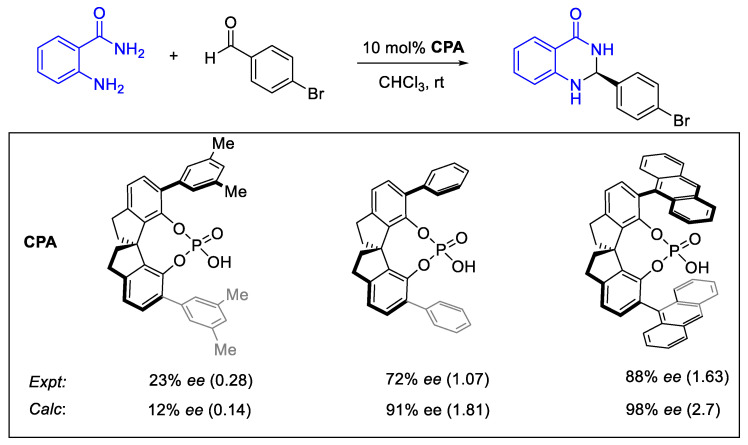
SPINOL-CPA catalyzed asymmetric synthesis of dihydroquinazolinones [186]. All values are obtained from the work of Wheeler and co-workers. % *ee* are calculated from ΔΔG^‡^_298.15 K, 1 atm_ at PCM(CHCl_3_)/B97-D3/def2-TZVP//PCM(CHCl_3_)/B97-D/def2-TZVP from the cited reference. Numbers in parentheses are the corresponding ΔΔG^‡^ at 298.15 K in kcal/mol.

**Figure 26 molecules-28-01715-f026:**
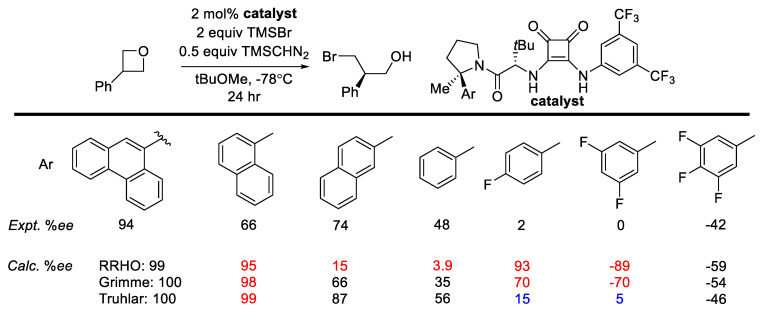
Squaramide-catalyzed enantioselective oxetane opening. Values are from the work of Jacobsen and co-workers [88]. Only the Lewis acid mechanism result is shown here. % *ee* are calculated from the ΔΔG^‡^_195.15 K, 1 atm_ calculated at SMD(Et_2_O)/B97D3/def2-TZVP//SMD(Et_2_O)/B97D/def2-SVP as reported by the cited reference.

**Figure 27 molecules-28-01715-f027:**
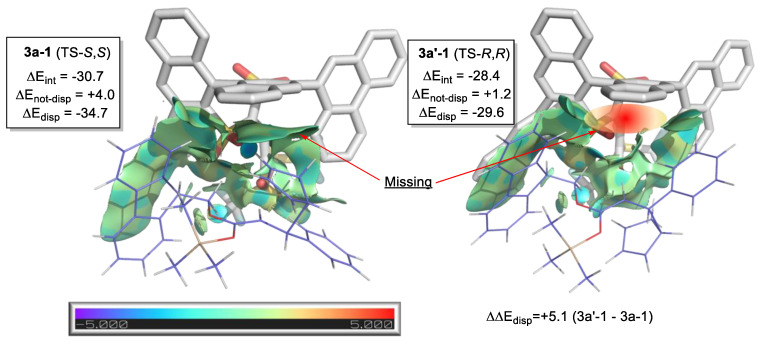
Geometries optimized at PBE-D3/def2-SVP were taken from Bistoni and co-workers [40]. NCI surface was generated by NCIplot with promolecular density. ΔE_int_ is the interaction energy between the catalyst and substrates calculated at DLPNO-CCSD(T)/def2-TZVP. ΔE_not-disp_ and ΔE_disp_ are the non-dispersion and dispersion contributions to the interaction energy, respectively, from LED analysis. All values are in kcal/mol and taken from Bistoni and co-workers [40].

**Figure 28 molecules-28-01715-f028:**
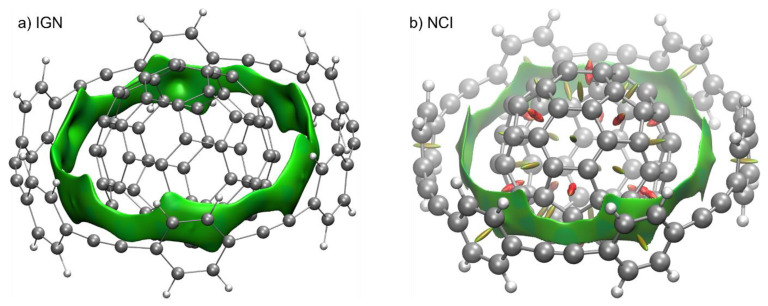
(**a**) IGM isosurface and (**b**) NCI isosurface generated with densities at B3LYP/6-31G(d). Visualization was done with VMD [196].

**Figure 29 molecules-28-01715-f029:**
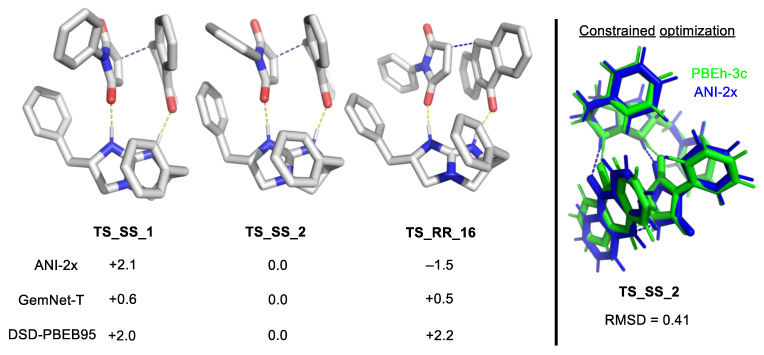
DSD-PBEB95: DSD-PBEB95-D3(BJ)/def2-QZVP. Values are taken from ref [32]. Geometries are generated from coordinates supplied by ref [32]. RMSD calculated by alignment function of PYMOL. Constrained optimization is done with ASE. In the constrained optimization, the two C atoms involved in the C-C bond formation are fixed (atom constraint).

**Table 2 molecules-28-01715-t002:** Energetic span and TOF at 223.15 K and 298.15 K.

Temperature	Energetic Span (kcal/mol) *	TOF (h^−1^) *
**298.15**	20	49
	21	9.1
	22	1.67
**223.15**	15	34
	16	3.58
	17	0.38

* Calculated with AUTOF_Excel_v09_1 [100].

**Table 3 molecules-28-01715-t003:** The relative computational cost of selected DFA from Table 1.

DFA	Relative Cost of Energy Calculation ^[a]^*	Relative Cost of Gradient Calculation *
revPBE	1.0	1.0
r^2^SCAN	1.1	1.0
M06-2X	5.5	4.5
ωB97M-V	3.3	3.0 ^[b]^
revDSD-PBEP86	360 ^[c]^	510 ^[c]^

TS_SS_2 from ref [32] is used. * All calculations are performed with ORCA 5.0.3 [66]. Variation is expected depending on system size and hardware setup. The following simple keywords: “def2-TZVPP autoaux verytightscf” are used. ^[a]^ Relative cost is estimated from the total SCF time reported by ORCA 5.0.3 for the SCF cycle part and divided by the number of iterations taken to converge the SCF calculation. This is normalized with the DFA with the minimum time taken—revPBE. ^[b]^ Including the time taken to calculate the non-local dispersion correction gradient. ^[c]^ Including DLPNO-MP2 calculation time.

**Table 4 molecules-28-01715-t004:** The volume of a cube to contain one particle at various concentrations and the number of water molecules to fill the same cube.

Reactant Concentration (M)	Vol. of a Cube to Contain One Molecule (Å^3^) ^[a]^	Number of H_2_O Molecules in the Volume Given on the Left ^[b]^
1 M	1660 (11.84)	56
0.1 M	16,605 (25.512)	555
0.01 M	166,053 (54.965)	5550

^[a]^ Conversion of M to particle/Å^3^ = 6.022 × 10^−4^. The length of the cube in Å is given in parentheses. ^[b]^ Based on a density of 1 g/mL and molar mass of 18 g/mol.

## Data Availability

The data presented in this study are openly available in FigShare at https://doi.org/10.6084/m9.figshare.22004126.

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
