# Peer review of "Molecular Understanding and Practical In Silico Catalyst Design in Computational Organocatalysis and Phase Transfer Catalysis—Challenges and Opportunities"

_molecules, 2023, doi:10.3390/molecules28041715_

Round 1

Reviewer 1 Report

The manuscript entitled “Molecular understanding and practical in silico catalyst design in computational organocatalysis and phase transfer catalysis Challenges and Opportunities. Therefore, I recommend this work could be published after the major revision

1.      Should the author write down the novelty of this review article in the abstract?

2.      The English composition requires many improvements. The authors should proofread the manuscript carefully to minimize grammatical errors.

3.      It would be better to add a schematic diagram that will summarize the review article.

4.      All the references mentioned in the paper should be cited in the text or vice-versa.

5.      This research topic based on electrospun fibers with tunable electrical conductive-semiconductive properties has been widely studied. The author, please add a comparative table for the reader's clear understanding.

6.      Please correct the typo error in the introduction part i.e. 1 in Erro and so on

7.      Please correct the error in Geometry optimization i.e. Error! Reference source not found and so on.

8.      Increase the font size of Figure 4 for readers to clear view. 

Reviewer 2 Report

This manuscript describes the key components to calculate or predict catalysis-performance metrics such as turnover frequency and measurement of stereoselectivity via computational chemistry in the field of organocatalysis and phase transfer catalysis. The work has been carried out with care and the results have been presented with clarity and discussed appropriately.  The topis is relevant and original en the field. The manuscript is a good review about the molecular understanding and practical in silico catalyst design in computational organocatalysis and phase transfer catalysis. In this manuscript, the state-of-the-art tools available to calculate potential energy and, consequently, free energy, together with their caveats, are discussed via examples from the literature, that validated their theoretical models. In this way, this manuscript demonstrates a significant contribution in the area of chemistry. The references are appropriate. The content of the manuscript is correct. The format of the text should be reviewed and the reference of the tables and figures in the text reviewed.
Accordingly, I can recommend publication of this work in Molecules after solving the issues detected.

Issues:

- Authors should check grammatical or formatting errors. ‘‘Error! Reference source not found.’’ is repeated many times in the text. Tables and Figures should be cited in the text. Words with different colors appear in the text and spaces between words.

- Authors should check and put the bibliography in the correct format.

- Authors should thoroughly review the text of the manuscript.

Reviewer 3 Report

This is very comprehensive review paper providing too much information on understanding, challenge and potential chances.  It can be accepted by smoothing the logic and style for easier reading. 

line 43-44, it is better to give a magnitude of time required, to see the  cost of balance with accuracy. 

section 2, as large pages review, it is better to give a content list at the beginning of manuscript, and number of subtitles in section 2. 

the abbreviation of special phase should be given in first appearance,  in the latter, abbr. is better, such as PES in line 141

line 223-224, please give more discussion on third rung 

line 525-527, as comparison, is it possible to provide the computing time or cost of each accurate method?

figure 12, reference 131 is not mentioned in the text. 

figure 15, lost A,B and C referring each figure.

figure 25, it is better to give more explanation on the difference between Expr. and Calc. in Figure 25, more than 10% absolute deviation. 

line 1163-1168, ML is potential research method, while, the data set is the base for the performance of ML, the precondition is the simulation results should be fully validated. This may be the more challenging work than ML please discuss this more. 

grammar and language

line 136, the condensed phase where organocatalysis

line 234, on DFA that barrier height

line 284, Based on the structures provided by Brandenburg, Tkatchenko, and co-workers.

line 813, between and , not with

Round 2

Reviewer 1 Report

The author solve all comments very carefully i recommended to accept in presence form.

Reviewer 2 Report

This manuscript describes the key components to calculate or predict catalysis-performance metrics such as turnover frequency and measurement of stereoselectivity via computational chemistry in the field of organocatalysis and phase transfer catalysis. The work has been carried out with care and the results have been presented with clarity and discussed appropriately. In this manuscript, the state-of-the-art tools available to calculate potential energy and, consequently, free energy, together with their caveats, are discussed via examples from the literature, that validated their theoretical models. In this way, this manuscript demonstrates a significant contribution in the area of chemistry. Accordingly, I can recommend publication of this work in Molecules. The authors have correctly performed the indicated revisions.